



# Control design for floating wind turbines

Amr Hegazy[1], Peter Naaijen[2], and Jan-Willem van Wingerden[1]

[1]Delft Center for Systems and Control, TU Delft, Mekelweg 2, 2628 CD, Delft, The Netherlands
[2]Maritime and Transport Technology, TU Delft, Mekelweg 2, 2628 CD, Delft, The Netherlands

**Correspondence:** Amr Hegazy (a.r.hegazy@tudelft.nl)

**Abstract.** While the feedback control of onshore wind turbines is well-established, applying the same controllers to floating offshore wind turbines causes the turbines to become unstable. Such instability is attributed to the coupling between the fore-aft motion and the wind turbine controller, which makes the wind turbine negatively damped. The non-minimum phase zeros existing in the transfer function from the blade pitch to the generator speed impose a fundamental limitation on the closed-loop bandwidth, posing a challenge to the operation of the floating turbines. This paper gives an overview of the control strategies and their tuning techniques employed for floating wind turbines in the presence of the negative damping instability. It discusses the different available strategies. Moreover, we propose a new controller that can alleviate the adverse effects of the negative damping while preserving the standard proportional-integral control structure. Contrary to the multi-input-multi-output controllers that have been proposed, the proposed controller is more robust since it does not require additional signals of the floating platform, which makes controllers often sensitive to unmodelled dynamics. The controller is compared against the previously proposed controllers using the non-linear simulation tool, OpenFAST. The proposed controller excels in regulating generator speed, surpassing other controllers in performance. Additionally, it effectively mitigates the platform pitch in addition to the tower and blade loads. However, achieving a balance between power quality, actuator usage, and structural loading presents inherent trade-offs that need to be carefully addressed.

## 1 Introduction

Wind energy is essential to meeting the decarbonisation objectives of the European Union (EU) energy system, as it ensures delivering clean, affordable and secure electricity to various sectors, including households, industry and transport. Consequently, wind energy is expected to heavily contribute to the EU renewable energy targets. This is not surprising, especially when we know that in 2024, wind energy covered $19\%$ of the EU electricity demand. No wonder the EU is regarded as a pioneer in wind energy. Accordingly, this has seen the EU revising the renewable energy directive, which lays down a minimum binding target of $42.5\%$ share of renewables by 2030 with an aspiration to reach $45\%$. This is $10.5\%$ higher than the initial $32\%$ target. Subsequently, the EU could fulfil its ambition of becoming climate-neutral by 2050 (European Commission, 2023).

As of 2025, Europe boasts a total installed wind capacity of approximately 285 gigawatts (GW), marking a significant expansion in the region's renewable energy infrastructure. Wind power accounts for almost $20\%$ of Europe's electricity consumption nowadays, and projections indicate that this figure could rise in the future. The EU aims to increase its wind capacity from 225 GW today to 350 GW by 2030, with a target of 425 GW to align with ambitious energy security goals (WindEurope, 2025).





Offshore wind offers significant advantages over onshore wind due to higher wind speeds and more consistent wind directions. Floating Offshore Wind Turbines (FOWTs) present unique opportunities as they can be deployed in deeper waters and farther from shore compared to bottom-fixed turbines. This expands the potential for offshore wind development in regions
with deeper sea basins, such as the Mediterranean and the Atlantic. However, FOWTs face harsher environmental conditions than onshore turbines. Unlike onshore turbines, FOWTs are subjected to additional disturbances caused by waves, which contribute to increased structural loading on top of the loads induced by wind turbulence. As a result, FOWTs experience higher fatigue loads due to the added impact of waves (Saenz-Aguirre et al., 2022).

The cost of energy defines the potential of one type of energy source to be preferred over another, with the Levelised Cost Of
Energy (LCOE) being the metric representing the average cost of generating electricity over the lifetime of a power-generating asset, expressed in monetary terms per unit of electricity. The main challenge facing the further deployment of FOWTs is their high LCOE. While modifications to their aerodynamic, hydrodynamic and structural design are applied to bring the LCOE down, the control system should not be overlooked as it can significantly contribute to reducing the LCOE.

From a control perspective, FOWTs present additional complexities compared to onshore turbines. The dynamics introduced
by the floating platform make control more challenging. A notable concern is the negative damping effect (Nielsen et al., 2006), as applying a fixed-bottom controller to a floating wind turbine can significantly amplify the system's dynamic response, leading to large peak-to-peak oscillations. The simplest way to avoid closed-loop instability without modifying the conventional baseline controller structure is to detune the control gains such that the closed-loop response of the generator speed mode in isolation has a natural frequency below the platform pitch resonant frequency (Larsen and Hanson, 2007; Jonkman, 2008).
However, this leads to a degradation in the reference tracking performance of the blade pitch controller as its ability to effectively respond to disturbances becomes restricted (Yu et al., 2018; Lemmer et al., 2020). Maintaining global detuning across all wind speeds sacrifices higher control bandwidths at higher wind speeds that do not suffer from this instability. Accordingly, it is reasonable to schedule the detuning at each wind speed separately (Yu et al., 2018, 2020; Lemmer et al., 2020; Stockhouse et al., 2024).
Other methods explored in the literature involve incorporating extra feedback loops to counteract the instability arising from rotor-platform interactions. By utilising nacelle fore-aft velocity as feedback to adjust the existing baseline controller actuators, blade pitch (Jonkman, 2008; van der Veen et al., 2012; Fleming et al., 2014) and generator torque (Fischer, 2013; Fischer and Loepelmann, 2016) control inputs showed performance improvements could be achieved without the need for additional actuators. A multi-loop feedback system evaluation requires a Multi-Input, Multi-Output (MIMO) transfer function representation.
Those multi-loop FOWT control strategies in the literature often employ a compartmentalised feedback design, where individual control channels are separately tuned to achieve improved dynamic responses of a specific output. While this segmented tuning methodology remains widespread, inter-loop dynamic coupling inherent in MIMO architectures generates cross-channel interference phenomena, whereby localised parameter adjustment in a single control loop perturbs the closed-loop response characteristics of adjacent feedback channels. It was demonstrated that improved performance could be achieved when opti-
mally tuning all the control loops collectively accounting for the interdependencies within the MIMO feedback structure rather than tuning each control loop independently (Stockhouse et al., 2024). Modern multivariable control methodologies employing

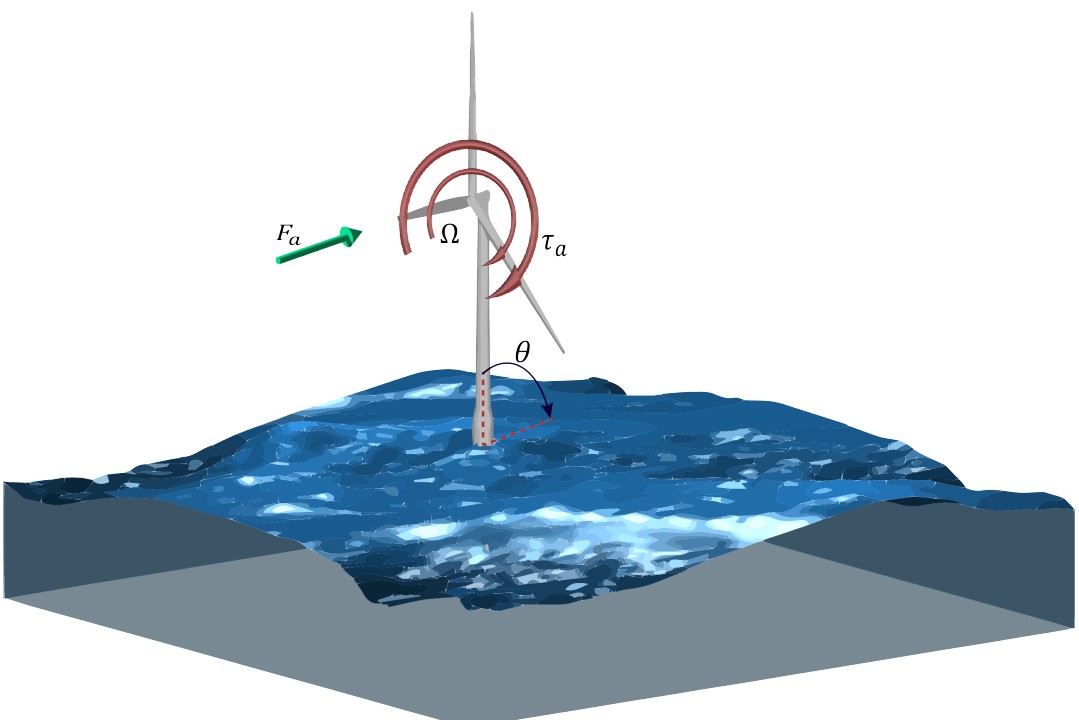

**Figure 1.** Schematic of the FOWT depicting the generator speed, $\Omega$, and the platform pitch, $\theta$, DoFs of the simple control model together with the external forces, namely, the aerodynamic thrust, $F_a$, and torque, $\tau_a$, as expressed in Eq. (14).

state-feedback architectures, including Linear Quadratic Regulator (LQR) (Namik et al., 2008) and $\mathcal{H}_\infty$ control (De Corcuera et al., 2012; Hegazy et al., 2023a) demonstrate systematic efficacy in achieving specified stability and performance envelopes for complex dynamical systems (Skogestad and Postlethwaite, 2005).

This paper provides a tutorial on the design of closed-loop controllers for FOWTs, outlining various control strategies proposed in the literature. It also evaluates the performance of these strategies, particularly in addressing the negative damping instability. Additionally, a novel controller structure is introduced, which eliminates the need for additional sensors, along with a detailed tuning technique.

The remainder of this paper is structured as follows: In Section 2, the FOWT control problem is defined, and the control
design model is introduced. In Section 3, conventional SISO and MIMO control strategies are discussed. In Section 4, the controllers are evaluated by simulating the closed-loop system using the non-linear higher-fidelity aero-servo-hydro-elastic tool OpenFAST (NREL, 2025a).





## 2 Problem background

In this section, we start by introducing the dynamic model of a conventional fixed-bottom wind turbine. Afterwards, we go on
to explain the process of closing the loop with the baseline controller and the tuning methodology of the controller gains. Once
this is established, we move on to the FOWTs where we introduce the additional dynamics for the floating platform to form a
representative dynamic model of a FOWT to conduct further analysis on the complexities that arise when controlling FOWTs.

Conventional wind turbine dynamics are excited by the imbalance between the aerodynamic torque and the generator torque,
which drives the generator speed, and thus a simplified wind turbine model is described as:

$$
\quad \frac{J_r}{N_{gb}}\dot{\Omega} = \tau_a(\Omega,\beta,v) - N_{gb}\tau_g, \tag{1}
$$

where $J_r$ (kg.m$^2$) is the rotor drivetrain inertia, $\Omega$ (rad.s$^{-1}$) is the generator speed with the dot notation indicating the time
derivative, $\tau_a$ (N.m) is the aerodynamic torque, $N_{gb}$ (-) is the gearbox ratio, and $\tau_g$ (N.m) is the generator torque. The aerody-
namic torque $\tau_a(\Omega,\beta,v)$ is modelled by a non-linear function:

$$
\tau_a = \frac{1}{2}\rho\pi R^2 N_{gb}\frac{v^3}{\Omega}C_p(\lambda,\beta), \tag{2}
$$

with $\rho$ (kg.m$^{-3}$) as the air density, $R$ (m) being the rotor radius, $C_p(\lambda,\beta)$ as the power coefficient, which depends on the
blade pitch angle, $\beta$ (rad), and the tip-speed ratio, $\lambda = \Omega R/v N_{gb}$, with $v$ (m.s$^{-1}$) being the wind speed normal to the rotor
plane. Around an equilibrium point, $\bar{x}$, a perturbation state, $\delta x$, is defined as $\delta x = x - \bar{x}$, and a non-linear function $f(\bar{x}) = 0$.
Therefore, the non-linear wind turbine dynamics in Eq. (1), at steady-state, can be linearised using first-order Taylor series
expansion around an equilibrium point as:

$$
\quad \delta\dot{\Omega} = \frac{N_{gb}}{J_r}\left(\frac{\partial\tau_a}{\partial\Omega}\delta\Omega + \frac{\partial\tau_a}{\partial v}\delta v + \frac{\partial\tau_a}{\partial\beta}\delta\beta - N_{gb}\delta\tau_g\right), \tag{3}
$$

where the partial derivatives of $\tau_a$ with respect to its independent variables are known as the aerodynamic sensitivities.

In practice, wind turbines are regulated with a generator speed controller, as at below-rated wind speeds, the controller is
seeking to maximise the extracted power by keeping the collective blade pitch angle, $\beta$, constant while varying the generator
torque, $\tau_g$, as a function of the square of the generator speed, $\Omega$, as follows (Bossanyi, 2000):

$$
\quad \tau_g = k_g\Omega^2, \tag{4}
$$

with $k_g = 0.5\rho\pi R^2(C_{p,max}/N_{gb}^3\lambda_{opt}^3)$, being the generator-torque constant. The variable $C_{p,max}$ is the maximum power
coefficient achieved at the optimal Tip-Speed Ratio (TSR), $\lambda_{opt}$, and at a specific constant blade pitch angle known as fine
blade pitch angle. Although simple, the controller in Eq. (4) operates under the assumption that $k_g$ remains constant throughout
the wind turbine's lifetime. In reality, this is not the case, as it is influenced by modelling inaccuracies and assumption errors.





To address this limitation, the TSR tracking controller has proven to be more effective and is widely adopted in the wind energy industry (Abbas et al., 2022; Brandetti et al., 2023).

At above-rated wind speeds (referred to as Region 3), a conventional wind turbine controller relies on the blade pitch to regulate the generator speed to its rated value while keeping the generator torque constant at its rated value (Bossanyi, 2000). As a result, generator power fluctuations are directly proportional to the oscillations occurring in the generator speed, $\delta\Omega = \Omega - \bar{\Omega}$. The collective blade pitch controller regulates the generator speed about its steady-state value, $\bar{\Omega} = \Omega_{rat}$, according to the following feedback control law:

$$\delta\beta = k_p \delta\Omega + k_i \int \delta\Omega dt, \tag{5}$$

where $k_p$ and $k_i$ are the proportional and integral controller gains, respectively. To reach a description of the gains, the azimuth angle $\psi$ is introduced as $\Omega = \dot{\psi}$ in Eq. (3) and Eq. (5). By combining both equations and focusing on the generator speed terms, we derive a closed-loop system. When rewritten in the standard form of a second-order mass-spring-damper system, it becomes:

$$\delta\ddot{\psi} + \underbrace{\frac{-N_{gb}}{J_r}\left(\frac{\partial\tau_a}{\partial\Omega} + \frac{\partial\tau_a}{\partial\beta}k_p\right)}_{2\zeta_c\omega_c}\delta\dot{\psi} + \underbrace{\frac{-N_{gb}}{J_r}\frac{\partial\tau_a}{\partial\beta}k_i}_{\omega_c^2}\delta\psi = 0. \tag{6}$$

Notice that the terms irrelevant to the control problem in Eq. (3) were dropped and do not appear in Eq. (6). Accordingly, we can parametrise the PI blade pitch controller gains:

$$k_i = -\omega_c^2 \frac{J_r}{N_{gb}}\left(\frac{\partial\tau_a}{\partial\beta}\right)^{-1} \tag{7}$$

$$k_p = \left(-2\zeta_c\omega_c\frac{J_r}{N_{gb}} - \frac{\partial\tau_a}{\partial\Omega}\right)\left(\frac{\partial\tau_a}{\partial\beta}\right)^{-1} \tag{8}$$

Given a desired natural frequency, $\omega_c$, and damping ratio, $\zeta_c$, the PI controller gains can be computed (Åström and Murray, 2021). By defining the $\omega_c$ and $\zeta_c$ of the generator speed response, the dynamic response of the rotor to wind speed variations can be altered. The value of $\omega_c$ defines the bandwidth of the feedback controller. Typically, the controller bandwidth is chosen below the lowest structural natural frequency of the system to avoid interaction with lightly damped modes, leading to instability. The bandwidth should not lie in the vicinity of the RHPZs existing in the wind turbine system as Leithead and Dominguez (2006) reported. As shown in Eq. (7) and Eq. (8), there controller gains depend on the aerodynamic sensitivities, which significantly vary across operating points. As a result, the controller gains are scheduled at each operating point and modified during operation as the wind speed changes to maintain consistent closed-loop transient behaviour using a linear controller.



The main challenge associated with the control of FOWTs, within Region 3 concerns their fore-aft motion (Larsen and Hanson, 2007; Jonkman, 2008; van der Veen et al., 2012; Fischer, 2013). Therefore, it is critical to include floating platform dynamics in the control design model.

## 2.1 Floating wind turbine model

The main problem associated with the control of floating wind turbines concerns the pitch stability in full load (van der Veen et al., 2012; Larsen and Hanson, 2007; Jonkman, 2008; Fischer, 2013). The effect of varying wind speed on the steady state thrust, in the above-rated region, has to be considered in order to understand this problem. The above-rated part of the steady state thrust curve, shown in Figure 2, is defined as the thrust force required at a given wind speed to produce rated power at rated generator speed. The steady-state blade pitch angle varies along the operating curve to achieve constant generator torque instead of constant power since this limits the generator speed variations, hence, reduced drive train loads and pitch activity (Larsen and Hanson, 2007). The objective is to achieve a stable power production with less variations such that its total differential diminishes.

To form a FOWT mathematical model, the generic 1-DOF model of the wind turbine in Eq. (3) is combined with the floating platform dynamics. For the sake of explaining the negative damping problem, only a 2-DOF FOWT model capturing the critical dynamics is used, where the platform pitch degree of freedom (DOF) is primarily considered to characterise platform dynamics, as the negative damping instability is most pronounced at the platform pitch eigenfrequency (Jonkman, 2008). However, to preserve key dynamic couplings, the control model used for the control design must include additional modes that capture the most significant system dynamics, namely the platform's surge and heave, and the tower first fore-aft bending (Lemmer et al., 2020); otherwise, some interactions within the system may be overlooked (Yu et al., 2020). The non-relevant DOFs are neglected to avoid accounting for extra states, which would increase the complexity.

The rigid floating platform pitch motion in still water, thus, affected by the aerodynamic thrust force only without any wave-induced forces, can be modelled as a second-order mass-spring-damper system:

$$I_p\ddot{\theta} + C\dot{\theta} + K\theta = l_h F_a(\Omega, v, \beta), \tag{9}$$

where $\theta$ is the platform pitch angle, $\dot{\theta}$ is the platform pitch rotational velocity, $\ddot{\theta}$ is the platform pitch rotational acceleration, $I_p$ is the total mass moment of inertia about the platform pitch axis, comprising the structural inertia and the added mass associated with hydrodynamic radiation, $C$ is the damping coefficient, $K$ includes the hydrostatic and the mooring stiffnesses. Within the simplified 2D model, the frequency-dependent radiation memory effects are disregarded by assuming a constant added mass and omitting radiation damping, as it is insignificant compared to viscous damping in FOWT platforms (Lemmer et al., 2016, 2020), while for the control model, a parametric radiation model is used (Perez and Fossen, 2009; Fontanella et al., 2020). However, for the time-domain simulations, the convolution integral (Cummins, 1961) is incorporated to account for the frequency-dependent coefficients. The variable $F_a$ is the aerodynamic rotor thrust force, which causes a pitching moment on the platform through the hub height, $l_h$, as a lever arm. The aerodynamic thrust force $F_a(\Omega, \beta, v)$ is a non-linear function is





expressed by:

$$F_a = \frac{1}{2}\rho\pi R^2 v^2 C_t(\lambda, \beta),\tag{10}$$

where $v$ is the rotor effective wind speed, $C_t$ is the thrust coefficient function in $\lambda$ and $\beta$. The platform pitch motion influences the dynamics as it induces a relative wind speed at the rotor apart from the inflow wind speed, $v_\infty$. Thus, the rotor effective wind speed, $v$, is:

$$v = v_\infty - l_h\dot{\theta}.\tag{11}$$

Similar to Eq. (3) while considering Eq. (11), the non-linear platform dynamics can be linearised around an equilibrium point as:

$$I_p\delta\ddot{\theta} + C\delta\dot{\theta} + K\delta\theta = l_h\left(\frac{\partial F_a}{\partial \Omega}\delta\Omega - l_h\frac{\partial F_a}{\partial v}\delta\dot{\theta} + \frac{\partial F_a}{\partial \beta}\delta\beta\right)\tag{12}$$

In a standard second-order form, by considering only the coefficients corresponding to the platform pitch motion, Eq. (12) can be rewritten as:

$$\delta\ddot{\theta} + \underbrace{\frac{1}{I_p}\left(C + l_h^2\frac{\partial F_a}{\partial v}\right)}_{2\zeta_p\omega_p}\delta\dot{\theta} + \underbrace{\frac{K}{I_p}}_{\omega_p^2}\delta\theta = 0,\tag{13}$$

with $\omega_p$ and $\zeta_p$ being the natural frequency and the damping ratio of the floating platform in the pitch DoF, respectively.

The coupled dynamics of the wind turbine in Eq. (3) and the floating platform in Eq. (12) form a third-order system, which is represented in state space form of $\dot{x} = Ax + Bu$, with a state vector $x = [\theta\,\dot{\theta}\,\Omega]^\top$, and control input vector $u = [\tau_g\,\beta]^\top$, as:

$$\dot{x} = \begin{bmatrix} 0 & 1 & 0 \\ A_K^\theta & A_C^\theta & A_\Omega^\theta \\ 0 & A_\theta^\Omega & A^\Omega \end{bmatrix} x + \begin{bmatrix} 0 & 0 \\ 0 & B_\beta^\theta \\ B_{\tau_g}^\Omega & B_\beta^\Omega \end{bmatrix} u,\tag{14}$$

where the individual elements of the system matrix, $A$, and the input matrix, $B$, are defined in Table 1. The output vector $y = Cx + Du$, with the output matrix $C$ and the feed-through matrix $D$, is defined according to the available system measurements, which is typically a subset of the states in the state vector $x$. In this paper, the output vector is chosen as $y = [\dot{\theta}\,\Omega]^\top$, and thus obtained for the state-space model in Eq. (14) as:





$$y = \begin{bmatrix} 0 & 1 & 0 \\ 0 & 0 & 1 \end{bmatrix} x, \tag{15}$$

The element $A_\theta^\Omega$ in the system matrix, $A$, in Eq. (14) is the state transition term from the platform pitch velocity, $\delta\dot\theta$, to the generator acceleration, $\delta\dot\omega_g$, which clearly shows the direct effect of the platform pitch motion on the generator acceleration via the term $\dfrac{-1}{J_r}\dfrac{\partial\tau_a}{\partial v}$.

**Table 1.** The elements of the system matrices $A$ and $B$.

| Element | Definition |
|---|---|
| $A_K^\theta$ | $-\dfrac{K}{I_p}$ |
| $A_C^\theta$ | $-\dfrac{1}{I_p}\left(C + l_h^2\dfrac{\partial F_a}{\partial v}\right)$ |
| $A_\Omega^\theta$ | $\dfrac{l_h}{I_p}\dfrac{\partial F_a}{\partial\Omega}$ |
| $A_\theta^\Omega$ | $-l_h\dfrac{N_{gb}}{J_r}\dfrac{\partial\tau_a}{\partial v}$ |
| $A^\Omega$ | $\dfrac{N_{gb}}{J_r}\dfrac{\partial\tau_a}{\partial\Omega}$ |
| $B_\beta^\theta$ | $\dfrac{l_h}{I_p}\dfrac{\partial F_a}{\partial\beta}$ |
| $B_{\tau_g}^\Omega$ | $-\dfrac{N_{gb}^2}{J_r}$ |
| $B_\beta^\Omega$ | $\dfrac{N_{gb}}{J_r}\dfrac{\partial\tau_a}{\partial\beta}$ |

Now with such linear state space model, we can view the problem analytically with a pole-zero plot, shown in Fig. 3, 

of the transfer function (TF), $G_{\Omega,\beta}$, mapping the collective pitch pitch, $\beta$, to generator speed, $\Omega$, describing how generator speed (controlled variable) responds to a variation in blade collective pitch angle (control input). First, let us look at the analytical description of $G_{\Omega,\beta}$. This requires transferring to the frequency domain, which can be attained by applying $G(s) =$





$C(s\boldsymbol{I} - \boldsymbol{A})^{-1}\boldsymbol{B} + \boldsymbol{D}$, with $s$ being the Laplace variable, and $\boldsymbol{I}$ being the identity matrix. As a result, we get a MIMO transfer function matrix, $\boldsymbol{G}(s) = \boldsymbol{G}_{\boldsymbol{uy}}(s)$ , mapping the input vector $\boldsymbol{u}$ to the output vector $\boldsymbol{y}$. The transfer function matrix $\boldsymbol{G}(s)$ is composed of SISO TF $G_{u_i y_i}(s) = y_i(s)/u_i(s)$ mapping each input $u_i(s)$ to each output $y_i(s)$:

$$\boldsymbol{G}(s) = \begin{bmatrix} G_{\dot{\theta},\tau_g} & G_{\dot{\theta},\beta} \\ G_{\Omega,\tau_g} & G_{\Omega,\beta} \end{bmatrix} \tag{16}$$

For the feedback control of FOWTs, the TF, $G_{\Omega,\beta}$ in Eq. (16) is of the main interest:

$$G_{\Omega,\beta} = \frac{\frac{\partial \tau_a}{\partial \beta}\left(I_p s^2 + \left[C + l_h^2 \frac{\partial F_a}{\partial v}\right] s + K\right) - l_h^2 \frac{\partial \tau_a}{\partial v} \frac{\partial F_a}{\partial \beta} s}{\left(\frac{J_r}{N_g b} s - \frac{\partial \tau_a}{\partial \Omega}\right)\left(I_p s^2 + \left[C + l_h^2 \frac{\partial F_a}{\partial v}\right] s + C\right) + l_h^2 \frac{\partial \tau_a}{\partial v} \frac{\partial F_a}{\partial \Omega} s}, \tag{17}$$

where all the gradients vary with the operating point. To determine the zeros of $G_{\Omega,\beta}$, its numerator polynomial is set to zero, and the resulting equation is solved for $s$ using the quadratic formula. Upon algebraic manipulation, it becomes evident that right-half plane zeros (RHPZs), indicating non-minimum phase behaviour, emerge under the following condition (Fischer, 2013):

$$C < -l_h^2 \underbrace{\left[\frac{\partial F_a}{\partial v} - \frac{\partial \tau_a}{\partial v}\frac{\partial F_a}{\partial \beta}\left(\frac{\partial \tau_a}{\partial \beta}\right)^{-1}\right]}_{\mu_{aero}} \tag{18}$$

Equation (18) highlights that the emergence of non-minimum phase behaviour, driven by the presence of right-half-plane zeros (RHPZs), is closely tied to the aerodynamic damping coefficient ($\mu_{aero}$), which is influenced by aerodynamic gradients. This coefficient varies with the operating conditions and tends to be particularly low near the rated wind speed, as will be demonstrated in the following analysis.

Figure 2 illustrates the relationship between the steady-state aerodynamic thrust force ($F_a$) and the rotor-effective wind speed ($v$) for above-rated operation of the NREL 5-MW reference wind turbine (Jonkman et al., 2009) installed on the OC3 spar floating platform (Jonkman, 2010). Under the assumption of quasi-static equilibrium, where system variables are balanced for each wind speed, the gradient $dF_a/dv$ is positive below-rated wind speed, meaning the thrust force increases as wind speed rises. However, beyond the rated wind speed, this gradient becomes negative, as shown in Fig. 2. This behaviour results from the pitch-to-feather control strategy, which reduces aerodynamic loads and modifies the force direction in the above-rated region. As a consequence, aerodynamic damping is positive at below-rated wind speeds but turns negative at above-rated wind speeds. As $F_a$ begins with a positive slope ($\mu_{aero} > 0$) in Region 2, where $F_a$ keeps increasing till reaching its maximum at the rated wind speed where $\mu_{aero} = 0$. Once Region 3 is reached, $F_a$ starts decreasing with a significantly steep negative slope ($\mu_{aero} < 0$). The steeper this decline, the lower the aerodynamic damping, with its minimum occurring just beyond the rated wind speed. As wind speed continues to increase, the slope gradually becomes less steep, indicating a partial recovery of aerodynamic damping.





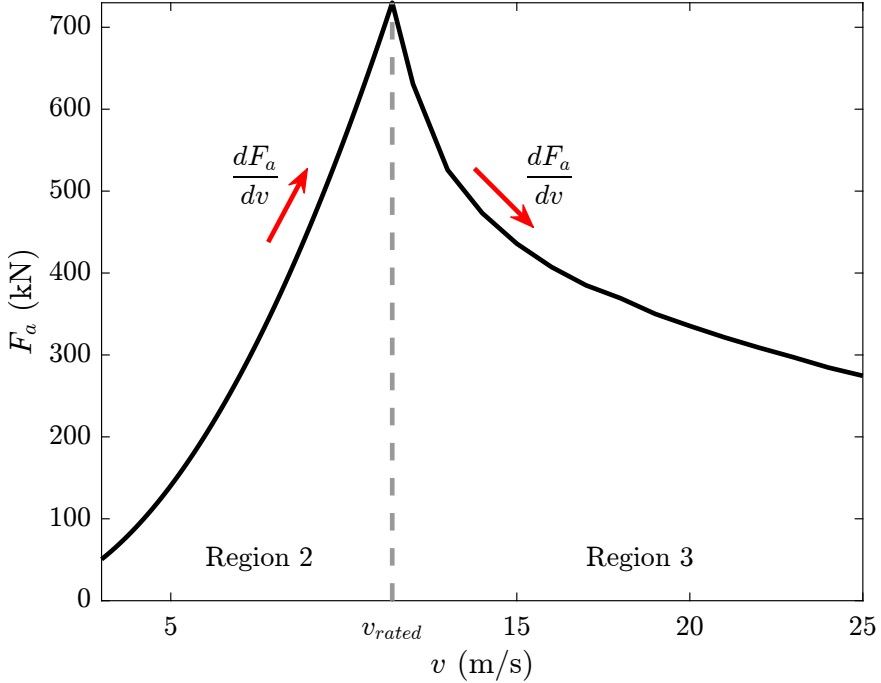

**Figure 2.** Steady-state values of rotor thrust force, $F_a$, as a function of the effective rotor wind speed, $v$, for the NREL 5-MW baseline wind turbine on OC3 spar floating platform.

The root cause of this behaviour is the negative total derivative of thrust force with respect to above-rated wind speeds (Fischer, 2013) as in Region 3, the rotor speed ($\Omega_r$) is at its constant rated value, while the aerodynamic torque ($\tau_a$) varies. The objective is to achieve stable power production ($P$) with fewer variations such that its total differential diminishes (van der Veen et al., 2012):

$$dP = \Omega_r d\tau_a = \Omega_r \left( \frac{\partial \tau_a}{\partial v}dv + \frac{\partial \tau_a}{\partial \beta}d\beta \right) = 0, \qquad (19)$$

and from Eq. (19), the total differential of the blade-pitch angle is:

$$d\beta = -\frac{\partial \tau_a}{\partial v} \left( \frac{\partial \tau_a}{\partial \beta} \right)^{-1} dv \qquad (20)$$

Similar to $d\tau_a$ in Eq.(19), the total differential of $F_a$ is:

$$dF_a = \frac{\partial F_a}{\partial v}dv + \frac{\partial F_a}{\partial \beta}d\beta \qquad (21)$$





Combining Eq. (20) and Eq. (21), the total derivative of the aerodynamic thrust with respect to the wind speed, yielded from

225 the variation of blade pitch to maintain rated power, is:

$$\frac{dF_a}{dv} = \frac{\partial F_a}{\partial v} - \frac{\partial F_a}{\partial \beta}\frac{\partial \tau_a}{\partial v}\left(\frac{\partial \tau_a}{\partial \beta}\right)^{-1} = \mu_{aero} \tag{22}$$

Equation (22) that demonstrates why $F_a$ has a negative gradient, $dF_a/dv < 0$, as wind speed increases, a condition that is

necessarily true for all conventional pitch-to-feather wind turbines (van der Veen et al., 2012). Burton et al. (2021) explains

that as the wind increases above-rated, the pitch angle increases to maintain constant generator torque, but the aerodynamic

thrust and torque decrease, indicating that the gradients $\partial F_a/\partial \beta$ and $\partial \tau_a/\partial \beta$ are negative. This allows the downwind fore-aft

motion to decrease, which leads to an upwind fore-aft motion, causing the relative wind speed seen by the rotor to increase.

Consequently, the aerodynamic torque increases further, causing more pitch action (Jonkman, 2008; van der Veen et al., 2012).

So, the gradient $\partial \tau_a/\partial v$ is positive. Therefore, after considering the signs of all the gradients in Eq. (22), it becomes clear why

$dF_a/dv < 0$ in the above-rated operation.

After obtaining $G_{\Omega,\beta}$ from $G(s)$ in Eq. (16), the pole-zero map of $G_{\Omega,\beta}$, that maps the blade collective pitch, $\beta$, to the

generator speed, $\Omega$, describing how the generator speed responds to a variation in blade pitch angle, is shown in Fig. 3.

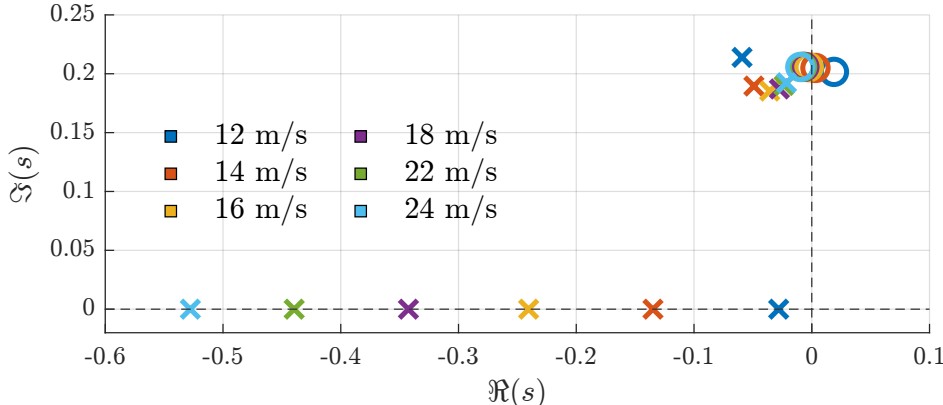

**Figure 3.** Pole-Zero map of the TF from blade collective pitch to rotor speed, $G_{\Omega,\beta}$, at different operating points. Poles and zeros are denoted by × and ∘, respectively.

Figure 3 shows that the TF, $G_{\Omega,\beta}$, consists of a complex pole pair, corresponding to the platform rigid-body pitch mode,

and a real pole, associated with the drivetrain mode. Additionally, a complex pair of right-half-plane-zeros (RHPZ) appears at

a frequency close to that of the platform pitch mode indicating that the RHPZs condition in Eq. (18) is satisfied. The poles in

the platform pitch mode of the open-loop transfer function, $G_{\Omega,\beta}$, correspond to the pitch-free decay damping ratio, $\zeta_p$, and

natural frequency (eigenfrequency), $\omega_p$. It can be seen in Fig. 3 that the open-loop system is originally stable because of the

sufficient hydrodynamic damping (Yu et al., 2018) since all the poles are in the left-half-plane (LHP).





However, the closed-loop poles of a system would migrate from the open-loop poles location towards the open-loop zeros as the feedback gain increases (van der Veen et al., 2012). Hence, according to Fig. 3, the platform pitch mode becomes less damped, whilst the generator speed tracking improves. In the case where the zeros are in the right half plane, which for the model visualized in Fig. 3 is true only for the platform pitch zeros, the frequencies provide bandwidth limits on $G_{\Omega,\beta}$ loop.

## 2.2 Effect of RHP zeros

A zero represents a critical frequency, referred to as the frequency of the zero, where the input signal is entirely blocked and has no effect on the system's output. In particular, right-half-plane zeros (RHPZs) exhibit an "inverse-response behavior," meaning the system output initially moves in the opposite direction of the expected response (Skogestad and Postlethwaite, 2005). This unique characteristic imposes strict constraints on control system design, especially in single-input single-output (SISO) configurations (Lemmer et al., 2016). Additionally, when the system is excited at or near the frequency of the zero, the risk of instability increases significantly. To mitigate this, limiting the controller bandwidth below the smallest RHPZ frequency is a must (Skogestad and Postlethwaite, 2005).

The effects of RHPZs extend beyond simple instability risks. As detailed in (Doyle et al., 2013), RHPZs introduce phase loss, which diminishes the performance of closed-loop systems as the zero frequency approaches the loop's cross-over frequency. This degradation becomes more critical in systems with weakly damped zeros (characterized by low damping ratios, $\zeta$), where abrupt phase shifts occur near the zero frequency, $\omega_z$. Such phase shifts are particularly problematic when the RHPZ frequencies fall below the controller bandwidth or the loop transfer function's cross-over frequency, exacerbating instability risks and limiting achievable performance. From a control design standpoint, RHPZs are universally undesirable due to their adverse impact on system stability and the fundamental limitations they impose on the achievable closed-loop bandwidth. Therefore, a careful balance between system performance and the trade-offs introduced by RHPZs should be considered, ensuring that controller bandwidth is appropriately tuned to account for these limitations.

## 3 Control of floating wind turbines

This section reviews various control strategies proposed for mitigating the negative damping instability in FOWTs, beginning with the most straightforward approaches and progressing toward more complex solutions involving additional sensors and actuators. Each method is evaluated in terms of its ability to address the negative damping effect and its effectiveness in overcoming the bandwidth limitation imposed by the RHPZs. Ultimately, the analysis concludes that only the incorporation of an additional actuator can effectively alleviate the constraint on closed-loop bandwidth—a point that is elaborated further in this section.





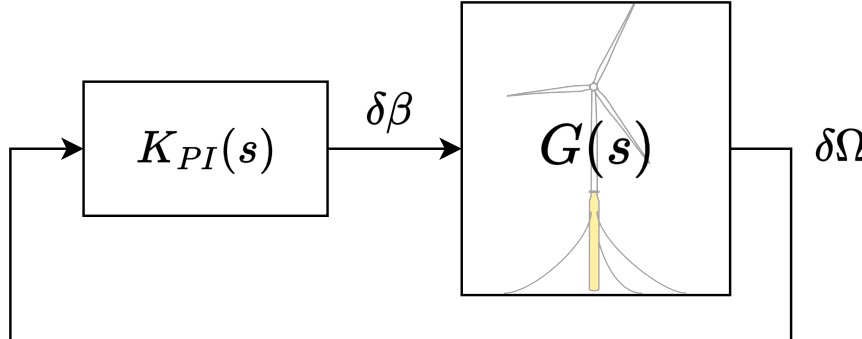

**Figure 4.** Block diagram of the FOWT closed loop system, where $G(s)$ represents the plant model, and $K_{PI}(s)$ represents the collective blade pitch controller.

Figure 4 shows the block diagram of the closed-loop FOWT system with the simple feedback PI controller. Each block represents a linear TF, with $G(s)$ mapping $\beta$, the collective blade pitch angle, to $\Omega$, the generator speed [rad/s], while $K_{PI}(s)$
is the collective blade pitch controller.

Neglecting the floating platform dynamics during the FOWT control design often yields instability in the operating points containing RHPZs. This is because of the high control bandwidth, triggered by the high feedback control gains, causing platform pitch excitation (Jonkman, 2008). At first, one might expect exponential growth in the response due to negative damping, but this is not the case because of the non-linear dynamic coupling between the different FOWT modes. Yet the FOWT keeps oscillating back and forth without reaching a steady state, which is still undesirable. There are several ways to mitigate this challenging problem. Thus, in the remainder of this section, the conventional solutions are presented, followed by our proposed solution in the next section.

### 3.1 Detuning

A common approach to mitigating negative damping instability is to reduce the bandwidth of the blade pitch controller below the platform's natural frequency (Larsen and Hanson, 2007; Jonkman, 2008; van der Veen et al., 2012). While this stabilises the system, it compromises generator speed tracking performance at operating points where detuning is implemented.

Detuning introduces a control performance trade-off in the vicinity of rated wind speeds. Lowering the closed-loop bandwidth to maintain stability compromises the system's disturbance rejection capability and degrades power tracking performance.

### 3.2 Robust scheduled tuning

As previously mentioned, stability can be maintained in the presence of RHPZs by detuning, such that the natural frequency of the closed-loop is below the frequency of the RHPZs, which is approximately equal to the resonant frequency of the platform pitch (Lemmer et al., 2020). Applying this approach means that the bandwidth and the damping ratio are constant across all



the operating points, which is inefficient since it sacrifices better tracking performance. According to Fig. 2 and Fig. 3, the limitation set by the RHPZs varies according to the operating point.

Instead of the global detuning explained in the previous section, a more efficient approach is to detune the PI controller to the fastest possible response at each operating point separately while maintaining the stability of the linear system (Lemmer et al., 2020; Yu et al., 2020; Stockhouse et al., 2024). In practice, it is not enough that a system is stable. There must also be some margins of stability that describe how far from instability the linear system is and its robustness to perturbations. The gain and phase margins are classical robustness measures that have been used for a long time in control system design, but they are not always good robustness indicators when it comes to the Nyquist stability criterion. However, the stability margin $s_m$ can be used instead to give a more general robustness measure. On one hand, it unites both the gain and phase margins under a single parameter, while on the other hand, it ensures that the Nyquist stability criterion is met. The stability margin $s_m$ is also a good robustness measure of nominally stable systems against model uncertainties. The stability margin of a closed-loop system is defined as the shortest distance between the Nyquist curve of the system's loop transfer function, $L(s) = G(s)K(s)$, and the critical point at $s = -1$ in the s-plane, and it expresses how well the Nyquist curve of the loop transfer avoids the critical point. While there is no representation of $s_m$ in the Bode plot of the loop transfer function, $s_m$ is related to the the peak magnitude, $M_s$, of the sensitivity closed-loop transfer function, $S(s) = (1 + L(s))^{-1}$, through $s_m = 1/M_s$, and $M_s$ being the $\mathcal{H}_\infty$ norm of $S(s)$ as (Åström and Murray, 2021):

$$s_m = \frac{1}{M_s} = \frac{1}{\|S(s)\|_\infty} \tag{23}$$

System stability robustness is a critical design priority for FOWTs, often leveraged in prior studies to calibrate both SISO (Lemmer et al., 2020) and MIMO control architectures (Stockhouse et al., 2024). The contour plots in Fig. 5 and Fig.6 depict the stability margin and the closed-loop bandwidth evaluated over a range of the proportional-integral (PI) control parameters, namely, $\omega_c$ and $\zeta_c$, showcasing the stable design space of the controller parameters, with the white-coloured region determining the unstable region. The stable region becomes larger as wind speed increases and the effect of the RHPZs fades according to Fig. 3, which allows for more freedom to increase the controller gains, and thus increase the closed-loop bandwidth without destabilising the system. It is important to mention that a stable design space means that the combination of the control parameters means a stable closed-loop system (i.e. not having right-half plane poles). Although the stable design space is extended at higher wind speeds, some combinations of the controller parameters would significantly increase the controller aggressiveness leading to failure in the non-linear simulations.

Increasing the closed-loop bandwidth reduces the stability margin, pushing the system closer to instability, as shown in Fig. 5 and Fig. 6. Consequently, achieving robust tuning of the PI controller requires a trade-off between stability robustness and closed-loop bandwidth, as these are competing objectives. An optimisation-based tuning integrating the two key system properties: the stability margin and the closed-loop system bandwidth while considering the actuator limits, is thus employed. The PI controller is parametrised by $\omega_p$ and $\zeta_c$ collected in the vector $\mathbf{x} \in \mathbb{R}^2$. A scalar objective function $J(\mathbf{x}) : \mathbb{R}^2 \to \mathbb{R}$ is

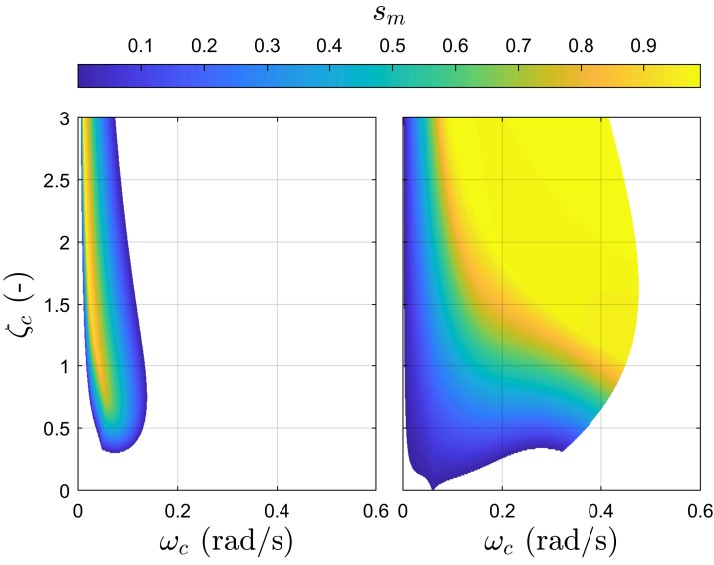

**Figure 5.** Stability margin contours across the natural frequency $\omega_c$ and damping ratio $\zeta_c$ of the PI controller, shown at two different operating points; near-rated ($\bar{v}$ = 13 m/s) and near cut-out ($\bar{v}$ = 24 m/s) wind speeds. The white region indicates a destabilising combination of $\omega_c$ and $\zeta_c$.

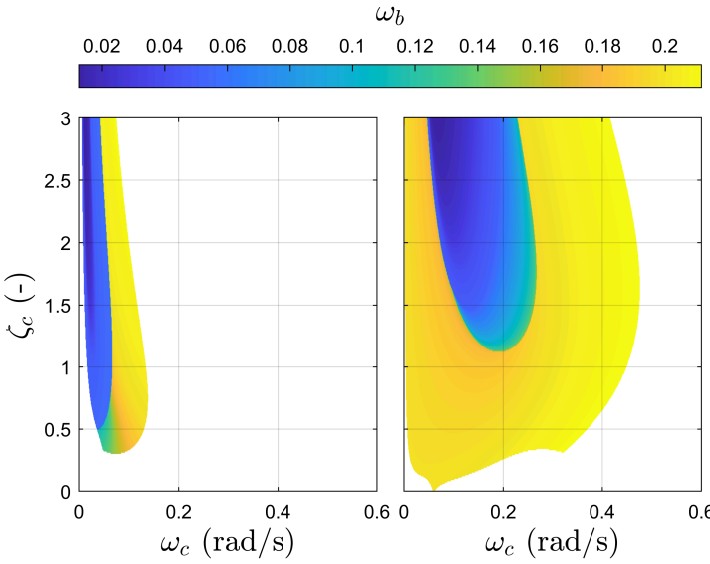

**Figure 6.** Closed-loop bandwidth contours across the natural frequency $\omega_c$ and damping ratio $\zeta_c$ of the PI controller, shown at two different operating points; near-rated ($\bar{v}$ = 13 m/s) and near cut-out ($\bar{v}$ = 24 m/s) wind speeds. The white region indicates a destabilising combination of $\omega_c$ and $\zeta_c$.



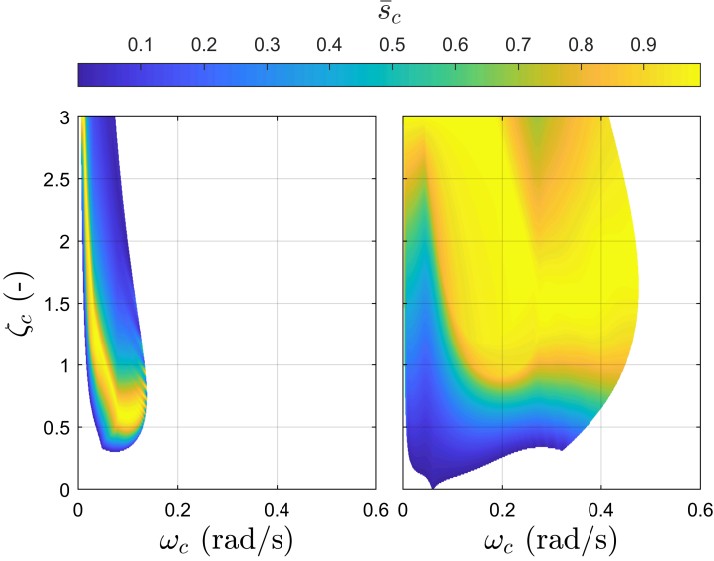

**Figure 7.** Control effort margin contours across the natural frequency $\omega_c$ and damping ratio $\zeta_c$ of the PI controller, shown at two different operating points; near-rated ($\bar{v}$ = 13 m/s) and near cut-out ($\bar{v}$ = 24 m/s) wind speeds. The white region indicates a destabilising combination of $\omega_c$ and $\zeta_c$.

then constructed with the following requirements: (i) maximise robust stability margin, (ii) maximise closed-loop bandwidth, and (iii) maintain acceptable actuator activity. When formulating $J(\mathbf{x})$, an important aspect is considering the actuator activity to avoid saturation. The control sensitivity function, $K(s)S(s)$, is a good indicator of the actuator activity. Inspired by $s_m$, 

the control effort margin $s_c$ is introduced here as a measure of actuation robustness. A low $s_c$ indicates high sensitivity to disturbances, risking actuator saturation. Analogous to $s_m$, we propose the variable $s_c$ that is related to the peak magnitude of the control sensitivity function $M_c$ through $s_c = 1/M_c$, where $M_c$ is defined as $M_c = \|K(s)S(s)\|_\infty$. The objective function is then formulated as:

$$J(\mathbf{x}) = \mathrm{w}_{s_m} s_m(\mathbf{x})^{-1} - \mathrm{w}_{bw} \omega_b(\mathbf{x}) + \mathrm{w}_{s_c} s_c(\mathbf{x})^{-1}, \tag{24}$$

where $\mathrm{w}_{s_m}$, $\mathrm{w}_{bw}$, and $\mathrm{w}_{s_c}$ are weights adjusting the importance of the stability margin, the bandwidth, and the control effort margin, respectively. Despite acknowledging that regularisation terms may be added to limit the gains, Stockhouse and Pao (2024) do not explicitly integrate actuator limits within their objective function formulation. Neglecting the actuator limits in the objective function would result in controller saturation. Conversely, Eq. (24) explicitly incorporates this constraint, ensuring the controller remains within operational limits. The objective function in Eq.(24) is then implemented in the optimisation

problem in the form:





$$\mathbf{x} = \underset{\mathbf{x}}{\operatorname{argmin}} \, J(\mathbf{x}) \tag{25}$$

In this framework, the optimisation variables (denoted as $\mathbf{x}$) are the tuning parameters influencing three critical system properties: the stability margin, the closed-loop bandwidth, and the control effort margin. A systematic tuning method, leveraging the simplified dynamic system, enables rapid recalibration of control settings and assessment of steady-state behaviour. The

core objective is to maximise the closed-loop bandwidth while minimising the inverse of the stability margin. Focusing on the inverse of the stability margin ensures the closed-loop stability of the system, while parameters that cause instability are dropped out. After formulating and weighting the objective function, a locally optimal solution is derived using a gradient-based optimisation solver.

Based on Eq. (24) and according to Fig.5 and Fig. 6, we have two competing objectives, as an increase in the closed-loop

bandwidth leads to a reduction in the closed-loop stability margin. Therefore, tuning the PI controller gains to achieve both objectives is not trivial, especially since finding a globally optimal solution is not guaranteed with gradient-based optimisation. Accordingly, a multi-objective optimisation problem is formulated over a set of continuous input variables $\mathcal{X} \subset \mathbb{R}^d$ called the $d$-dimensional design space (Lukovic et al., 2020). The optimisation goal is to maximise both the stability margin and the closed-loop bandwidth. The optimisation goal is to minimise the vector of the objectives defined as $f(x) = [f_1(x), \cdots, f_n(x)]$

with $n \geq 2$, $x \in \mathcal{X}$ being the vector of input variables and $f(\mathcal{X}) \subset \mathbb{R}^n$ the m-dimensional image representing the performance space.

The conflicting nature of the objectives does not always allow for the finding of a single optimal solution to the maximisation problem but a set of optimal solutions as shown in Fig. 8, referred to as the Pareto set $\mathcal{P}_s \subseteq \mathcal{X}$ in the design space and the Pareto front $\mathcal{P}_f = f(\mathcal{P}_s) \subset \mathbb{R}^n$ in the performance space (Lukovic et al., 2020).





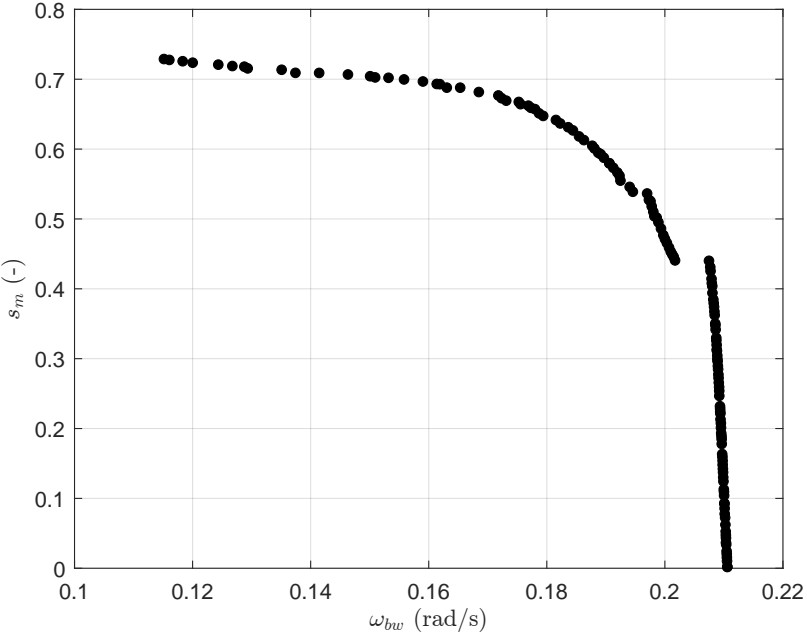

**Figure 8.** The Pareto front resulting from the multi-objective optimisation. Each data point indicates an optimal combination of the PI controller parameters $\omega_c$ and $\zeta_c$.

The Pareto front in Fig. 8 clearly illustrates the trade-off between closed-loop stability and bandwidth. Beyond a certain threshold, further increasing the bandwidth significantly compromises system stability. The knee point on the Pareto front represents an optimal balance between these competing objectives, making it a favorable region for selection. However, caution is needed when considering solutions in the upper-right region of the Pareto front. While they offer higher bandwidth, they also lead to excessive pitch activity, rendering them impractical due to actuator constraints.

### 3.3 Multi-loop control

A standard method to address negative damping instability involves implementing a secondary feedback loop that incorporates the platform pitch velocity signal. This technique can utilise blade pitch actuators (van der Veen et al., 2012) or generator torque actuators (Fischer, 2013), representing a shift toward MIMO control strategies. The approach seeks to reduce the coupling between competing aerodynamic forces—rotor torque and thrust—while maintaining generator speed regulation via blade pitch adjustments. In this work, the platform pitch rate is employed as the fore-aft velocity signal for the secondary feedback loop. The study evaluates both blade pitch damping and generator torque for parallel compensation, finding that combining the two actuators balances their advantages and limitations.

In Eq. (14) of the state-space model, the matrix element $A_\theta^\Omega$ represents the dynamic coupling between platform pitch velocity, $\dot{\theta}$, and rotor acceleration, $\dot{\Omega}$. Nullifying this term diminishes the influence of platform pitching on rotor speed tracking. This





tuning strategy does not directly suppress platform motion but counteracts its destabilizing effect on speed regulation, thereby enhancing closed-loop stability.

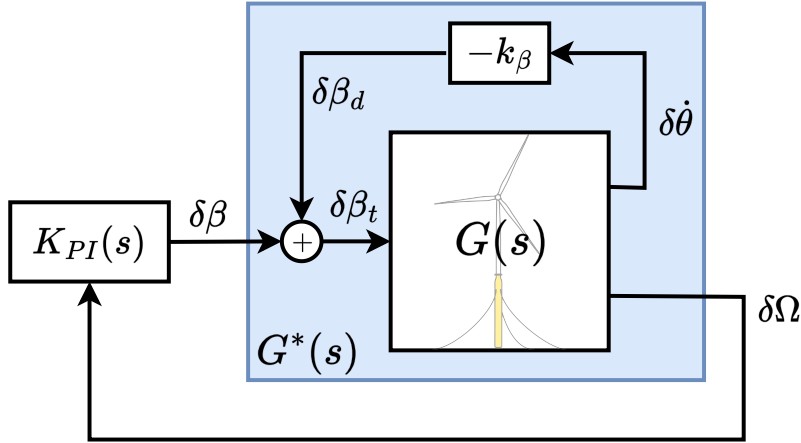

**Figure 9.** Block diagram of the blade pitch damping (MISO controller)

### 3.3.1 Blade pitch damping: MISO control structure

Compensation using blade pitch feedback, as shown in Fig. 9, is achieved by adding an extra term to the element $A_\theta^\Omega$, corresponding to the closure of the inner loop, where the static gain, $k_\beta$, is scheduled to be consistent with the PI controller gains for
each operating point. The blade pitch damping approach uses proportional feedback of the platform pitch velocity (Jonkman, 2008; van der Veen et al., 2012):

$$\delta\beta_d = -k_\beta\dot\theta \tag{26}$$

Therefore, the overall blade pitch signal becomes:

$$\delta\beta_t = \delta\beta + \delta\beta_d \tag{27}$$

Closing the inner feedback blade pitch loop by substituting $\delta\beta$ from Eq. (27) in Eq. (3) and Eq. (12), the system matrix of the inner loop, $A^*$, becomes:

$$
\boldsymbol{A^*} = \begin{bmatrix}
0 & 1 & 0 \\
-\dfrac{K}{I_p} & -\dfrac{1}{I_p}\left(C + l_h^2\dfrac{\partial F_a}{\partial v} + l_h k_\beta\dfrac{\partial F_a}{\partial \beta}\right) & \dfrac{l_h}{I_p}\dfrac{\partial F_a}{\partial \Omega} \\
0 & -l_h\dfrac{N_{gb}}{J_r}\left(\dfrac{\partial \tau_a}{\partial v} + k_\beta\dfrac{\partial \tau_a}{\partial \beta}\right) & \dfrac{N_{gb}}{J_r}\dfrac{\partial \tau_a}{\partial \Omega}
\end{bmatrix} \tag{28}
$$





This extra blade pitch in Eq. (26) is added to the collective blade pitch command from the PI controller, $K_{PI}(s)$ in Fig. 9, before the actuator saturation limits are applied. At first glance, it is observed that the extra feedback loop affects, not only the state transition from the platform pitch velocity to the generator speed, as shown by element $\boldsymbol{A}^*(3,2)$ but also the damping of the platform pitch mode shown by element $\boldsymbol{A}^*(2,2)$. This indicates that this parallel loop can be used for two control objectives; either to compensate for the RHPZs, or increase the platform pitch damping.

Solving for a gain that makes $\boldsymbol{A}^*(3,2) = 0$ leads to full compensation of the effect of platform pitch on the generator speed. However, due to blade pitch coupling with both aerodynamic torque and thrust, such a gain reduces the effective system fore-aft damping as a side effect. It is, therefore, sensible to choose a smaller gain to partially compensate the fore-aft motion, which can be achieved by multiplying the parallel compensation gain by a static gain, $\xi_\beta$. The parallel compensation gain for blade pitch then becomes

$$k_\beta = -\xi_\beta \frac{\partial \tau_a}{\partial v} \left( \frac{\partial \tau_a}{\partial \beta} \right)^{-1} \tag{29}$$

The value of $\xi_\beta \in [0,1]$ determines the degree of partial compensation from the blade pitch actuator to alleviate the effect of the platform pitch motion on the generator speed at the expense of less fore-aft damping. Should this objective be sought, extra filtering is required to change its dynamics, otherwise, it will be unstable. However, if the control objective shifts to increasing the fore-aft damping, that will be at the expense of reducing the drivetrain damping, thus resulting in less generator speed tracking performance. Similar to Eq. (13), the the platform pitch dynamics in the second row of $\boldsymbol{A}^*$ is represented in standard form as:

$$\delta\ddot{\theta} + \underbrace{\frac{1}{I_p} \left( C + l_h^2 \frac{\partial F_a}{\partial v} + l_h k_\beta \frac{\partial F_a}{\partial \beta} \right)}_{2\zeta_p^* \omega_p} \delta\dot{\theta} + \underbrace{\frac{K}{I_p}}_{\omega_p^2} \delta\theta = 0, \tag{30}$$

where $\zeta_p^*$ is the new desired damping ratio of the platform pitch DoF, without any change in its natural frequency. According to Eq. (30) and taking Eq. (13) into account, $k_\beta$ can be parametrised as:

$$k_\beta = \frac{2\omega_p \Delta\zeta_p}{\dfrac{l_h}{I_p} \dfrac{\partial F_a}{\partial \beta}} = \frac{2\omega_p(\zeta_p^* - \zeta_p)}{\dfrac{l_h}{I_p} \dfrac{\partial F_a}{\partial \beta}}, \tag{31}$$

where $\Delta\zeta_p$ represents the desired change in the platform damping. The extra feedback loop acts as a damper, increasing the system damping by moving the poles of $G_{\Omega,\beta}^*$, corresponding to the platform pitch mode, away from their respective zeros. While the RHPZs remain unaffected, setting restrictions on the closed-loop control performance, which is evident from the phase loss of $180°$ in Fig. 10, the damper effect is illustrated, highlighting its direct influence on the outer loop $G_{\Omega,\beta}^*$. It is

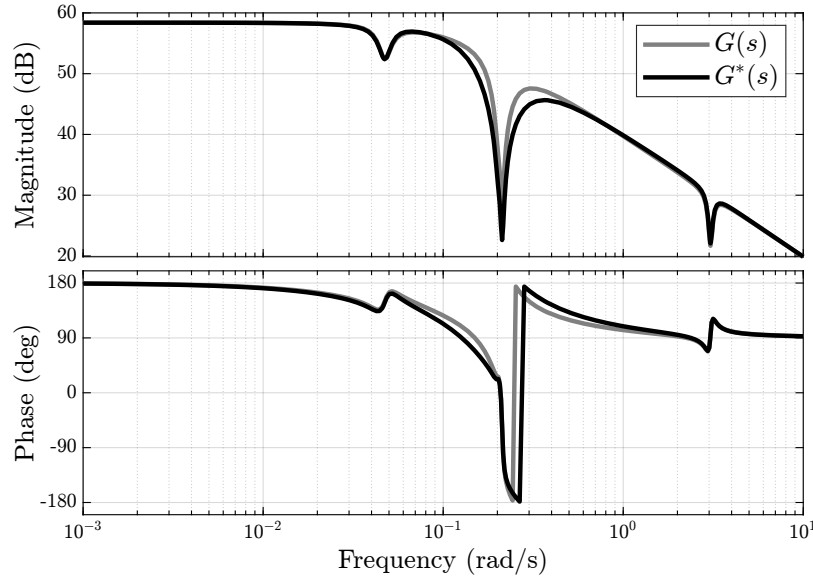

**Figure 10.** Bode plot comparing the channel mapping $\delta\beta$ to $\delta\Omega$ of the original transfer function $G(s)$ with the modified transfer function $G^*(s)$ depicted in Fig. 9, obtained after closing the inner loop from platform pitch velocity to blade pitch.

observed that the rotor dynamics deteriorate by adding the blade pitch damper as the depth of the anti-resonance dip increases, indicating an increase in generator speed oscillations, thereby affecting power production within the frequency range of the

fore-aft mode.

Although the MIMO plant, $G(s)$ does not have any transmission zeros, the poor generator speed tracking performance is attributed to the persistence of the RHPZs in $G^*_{\Omega,\beta}$, as they are not affected by the parallel inner loop, and still impose a limitation on the PI controller bandwidth. This is confirmed by checking the numerator of $G^*_{\Omega,\beta}$, whose damping term becomes:

$$C + l_h^2 \left( \underbrace{\frac{\partial F_a}{\partial v} - \frac{\partial \tau_a}{\partial v}\frac{\partial F_a}{\partial \beta}\left(\frac{\partial \tau_a}{\partial \beta}\right)^{-1}}_{\mu_{aero}} \right) - \cancel{k_\beta l_h \frac{\partial F_a}{\partial \beta}} + \cancel{k_\beta l_h \frac{\partial F_a}{\partial \beta}} \tag{32}$$

As shown in Eq. (32), the RHPZs are indeed unaffected since the inner-loop contribution cancels, thus leaving the RHPZs condition in Eq. (18) with no change.

### 3.3.2  Parallel compensation: MIMO control structure

So far, the previous control strategies proved not to be able to compensate for the deteriorating effect of the RHPZs. The only way to move zeros is by parallel compensation, $y = (G + K)u$, which, if $y$ is a physical output, can only be accomplished by

adding an extra input (actuator) (Skogestad and Postlethwaite, 2005).





As mentioned earlier, the presence of zeros implies the blockage of certain input signals. In this case, the blade pitch input is blocked due to the emergence of RHPZs, which is depicted in Fig. 10 where anti-resonance dips exist, indicating a significant attenuation of the input signals at those frequencies. Therefore, instead of using the blade pitch in the parallel loop, the generator torque can be used as illustrated in Fig. 11, thus taking a step towards MIMO control. Unlike the blade pitch, the

generator torque compensation is different as when $G_{\Omega,\beta}$, is closed with the generator torque parallel compensation loop, the RHPZs move to the LHP. At optimal gain, the RHPZs vanish from $G^*_{\Omega,\beta}$, which is the TF representing $G_{\Omega,\beta}$ after closing the generator torque parallel loop, indicating that the system became minimum phase. The generator torque parallel compensation uses proportional feedback of the platform pitch velocity (Fischer, 2013):

$$\delta\tau_g = -k_{\tau_g}\delta\dot{\theta} \tag{33}$$

Closing the inner feedback generator torque loop by substituting $\delta\tau_g$ from Eq. (33) in Eq. (3) and Eq. (12), the system matrix of the inner loop becomes:

$$\boldsymbol{A^*} = \begin{bmatrix} 0 & 1 & 0 \\ -\dfrac{K}{I_p} & -\dfrac{1}{I_p}\left(C + l_h^2\dfrac{\partial F_a}{\partial v}\right) & \dfrac{l_h}{I_p}\dfrac{\partial F_a}{\partial \Omega} \\ 0 & \dfrac{N_{gb}}{J_r}\left(k_{\tau_g}N_{gb} - l_h\dfrac{\partial \tau_a}{\partial v}\right) & \dfrac{N_{gb}}{J_r}\dfrac{\partial \tau_a}{\partial \Omega} \end{bmatrix} \tag{34}$$

Therefore, to eliminate the effect of platform pitch rate on the rotor dynamics, set $\boldsymbol{A^*}(3,2) = 0$. Consequently, the parallel compensation gain for the generator torque actuator is:

$$k_{\tau_g} = \xi_{\tau_g}\frac{l_h}{N_{gb}}\frac{\partial \tau_a}{\partial v}, \tag{35}$$

where $\xi_{\tau_g} \in [0,1]$ is introduced as a tunable parameter determining the intensity of parallel compensation since it is not necessary to remove the RHPZs totally. Having a glance at the numerator of $G^*_{\Omega,\beta}$, it can be noticed that adding the parallel compensation loop modifies the damping term in the numerator by modifying the aerodynamic coefficient, $\mu_{aero}$ in Eq. (18), to a new one, which in return, leads to a different zeros locations. The new aerodynamic coefficient becomes:

$$\tilde{\mu}_{aero} = \frac{\partial F_a}{\partial v} + (\xi_{\tau_g} - 1)\frac{\partial \tau_a}{\partial v}\frac{\partial F_a}{\partial \beta}\left(\frac{\partial \tau_a}{\partial \beta}\right)^{-1} \tag{36}$$

According to Eq. (36), the parallel compensation feedback loop makes it possible to manipulate the zeros of $G_{\Omega,\beta}$ and compensate for the RHPZs by pushing them towards the LHP (Fischer, 2013; Yu et al., 2018; Hegazy et al., 2023a; Stockhouse



et al., 2024). The level of compensation is tunable based on the tuning of the gain $\xi_{\tau_g}$. The higher $\xi_{\tau_g}$, the more the RHPZs move towards the LHP till they migrate to the LHP indicating the removal of those RHPZs. Consequently, the bandwidth of the

PI controller can be increased above the platform pitch mode. This is clear in Fig. 12, as the depth of the anti-resonance dip, corresponding to the RHPZs, decreases meaning that the limitation set by the RHPZs is vanishing, which gives the opportunity to increase the aggressiveness of the PI controller.

The main drawback of this approach is the generator torque limit for parallel compensation that can be supplied by the actuator. The usage of the full-compensation gain ($\xi_{\tau_g} = 1$) eliminates the RHPZs, thus, turning the system to minimum

phase for all operating points, however, the constraint imposed by the $\tau_g$ saturation restrains actuator signals exceeding the maximum generator torque. Reducing the compensation gain with $\xi_{\tau_g} \in [0,1]$ is rather advantageous in practice, as on one hand, it prohibits the generator torque actuator from saturating, and on the other hand, it reduces the drivetrain loads (Hegazy et al., 2023a). With $\xi_{\tau_g} < 1$, the RHPZs are partially compensated, allowing higher achievable bandwidth and, hence, improved performance.

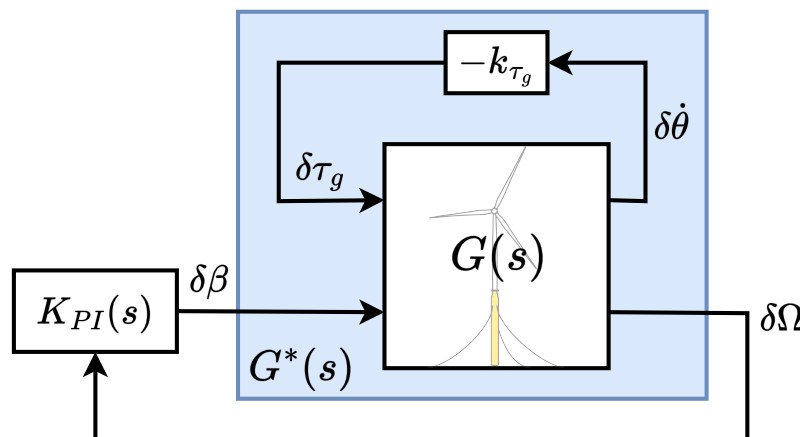

**Figure 11.** Block diagram of the generator torque parallel compensation (MIMO controller)

### 3.3.3   Parallel compensation: SIMO control structure

Hegazy et al. (2023a) showed that the feedback of the platform motion is not necessary for parallel compensation, as only generator speed can be used. They went on to show the control structure of the blade pitch and the generator torque controllers. It was learnt from $\mathcal{H}_\infty$ control synthesis that the blade pitch maintains the PI structure, while the generator torque requires a band-pass filter.

Figure 13 illustrates the control structure defined in Hegazy et al. (2023a), where the blade pitch controller maintains the PI control structure as:



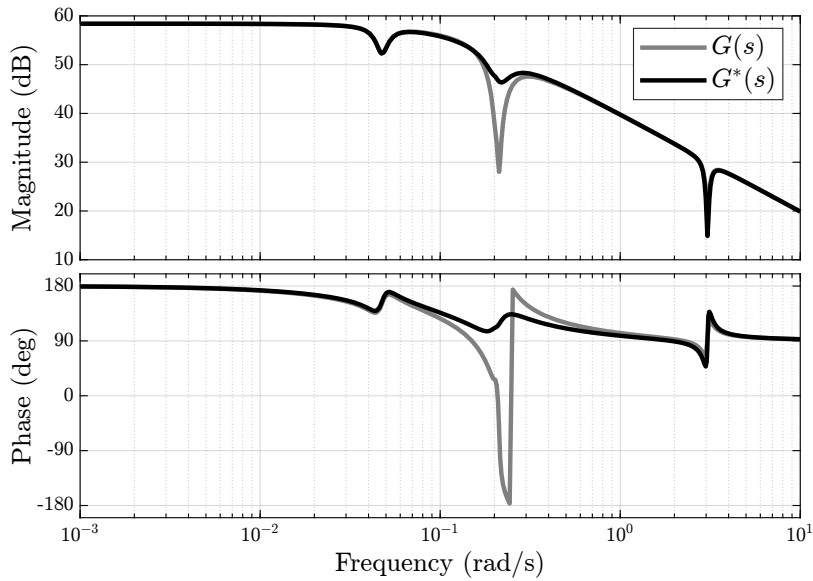

**Figure 12.** Bode plot comparing the original transfer function $G(s)$, which maps $\delta\beta$ to $\delta\Omega$, with the modified transfer function $G^*(s)$ depicted in Fig. 11, obtained after closing the parallel compensation feedback inner loop from platform pitch velocity to generator torque.

$$K_\beta(s) = k_p + \frac{k_i}{s},$$
(37)

where $k_p$ and $k_i$ are the proportional and the integral gains, respectively. As for the generator torque channel, an inverted notch is applied as:

$$K_{\tau_g}(s) = \frac{2\zeta_{\tau_g}\omega_{\tau_g}s}{s^2 + 2\zeta_{\tau_g}\omega_{\tau_g}s + \omega_{\tau_g}^2}$$
(38)

Consequently, the SIMO controller takes the form:

$$K(s) = \begin{bmatrix} K_{\tau_g}(s) \\ K_\beta(s) \end{bmatrix}$$
(39)





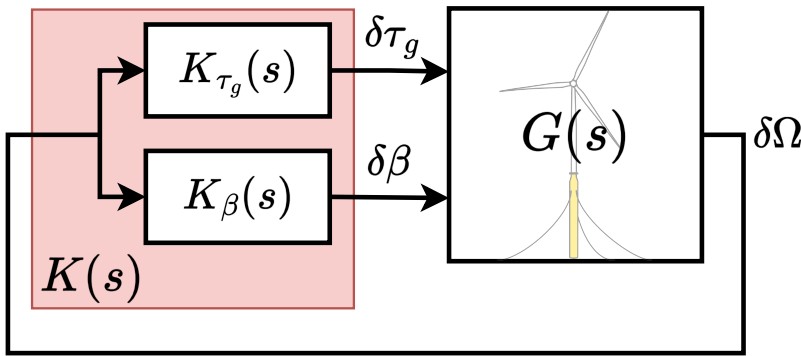

**Figure 13.** Block diagram of the FOWT closed-loop system, where $G(s)$ represents the plant model, and $K(s)$ represents the SIMO structure feedback controller composed of 2 SISO controllers; $K_{\tau_g}(s)$ controller acting on the generator torque actuator, and $K_\beta(s)$ active on the blade pitch actuator.

Now that the need for SIMO control to deal with the negative damping problem has been established in Fig. 13, tuning each controller separately sounds complicated due to the dynamic interactions between the MIMO channels that would arise when

either of the controllers is modified. Therefore, the objective is to turn the SIMO system into a SISO one. This is depicted in Fig. 14 where the extra blocks are integrated with the plant such that there is a new plant $G^*(s)$. This means that the new SISO plant $G^*(s)$ is the result of the linear combination of both control channels as:

$$G^*(s) = G(s) \begin{bmatrix} 1 \\ 1 \end{bmatrix} \tag{40}$$

In order to do that, the controllers $K_\beta(s)$ and $K_{\tau_g}(s)$ have to be decomposed such that:

$$K(s) = \begin{bmatrix} \tilde{K}_{\tau_g}(s) \\ \tilde{K}_\beta(s) \end{bmatrix} \tilde{K}(s) \tag{41}$$

where a band-pass filter is the outcome of combining a high-pass filter and an integrator:

$$K_{\tau_g}(s) = \tilde{K}(s)\tilde{K}_{\tau_g}(s) = \frac{2\zeta_{\tau_g}\omega_{\tau_g}\bar{k}}{s} \times \frac{s^2}{s^2 + 2\zeta_{\tau_g}\omega_{\tau_g}s + \omega_{\tau_g}^2}, \tag{42}$$

while a PI controller results from the combination of a PD and an integrator:

$$K_\beta(s) = \tilde{K}(s)\tilde{K}_\beta(s) = \frac{2\zeta_{\tau_g}\omega_{\tau_g}\bar{k}}{s} \times (\tilde{k}_p + \tilde{k}_d s), \tag{43}$$

where the PD controller gains are:





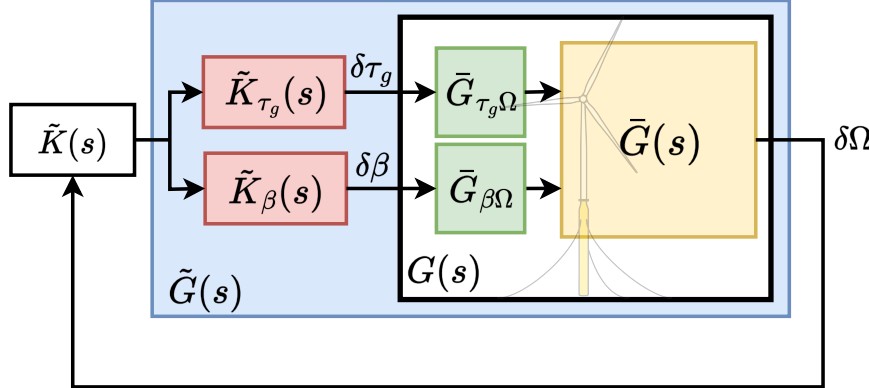

**Figure 14.** Transition from SIMO to SISO control structure

$$\tilde{k}_p = \frac{1}{2\zeta_{\tau_g}\omega_{\tau_g}} \tag{44}$$

$$\tilde{k}_d = \frac{1}{(2\zeta_{\tau_g}\omega_{\tau_g})^2} \tag{45}$$

The gain $\bar{k}$ in $\tilde{K}(s)$ is a static gain to either crank up or reduce the overall gain of the controllers $\tilde{K}_{\tau_g}$ and $\tilde{K}_\beta$ simultaneously. The objective is to tune one single controller instead of multiple control components, which would complicate the control tuning process.


The generator torque actuator is only active within the RHPZs frequency band to take over the control from the blade pitch, which is limited by the non-minimum phase behaviour around that band. This is depicted in Fig. 15, where the limitation set on the blade pitch, while regulating the generator speed, is lifted by the generator torque, and the linear combination of both actuators can lead to an increase in the control bandwidth, as shown in Fig. 16. The two vertical lines depict the closed-loop


bandwidth of each controller. Clearly, the baseline feedback PI controller has its bandwidth constrained by the RHPZs, which are also around the platform pitch natural frequency. Looking at the loop transfer function of the linear combination of both actuators, we can see the jump in the bandwidth the SIMO controller makes over the baseline controller, as the SIMO controller intersects with the 0 dB line much later than the baseline controller. Moreover, the anti-resonance dip that corresponds to the RHPZs existing in the bode plot of the baseline controller is eliminated in the SIMO controller, reflecting on its robustness as


it significantly increased with a phase margin of almost 90 deg.

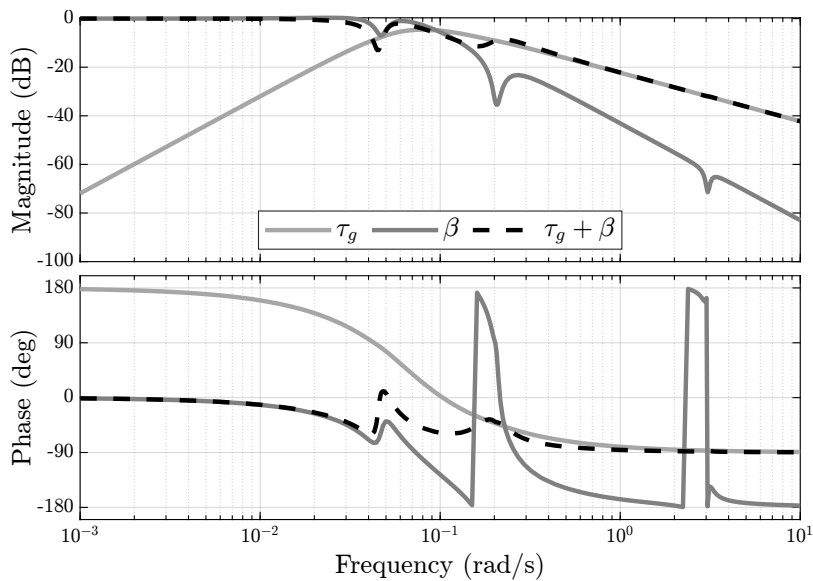

**Figure 15.** Bode plot illustrating the effect of the linear combination of both actuators where the blade pitch actuator is active till a certain frequency before its authority deteriorates, thus, the generator torque actuator takes over from that frequency onwards.

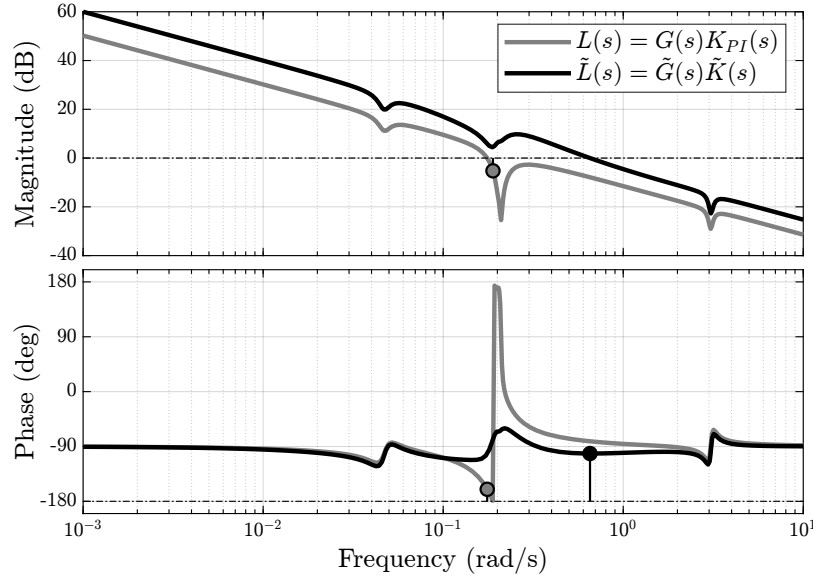

**Figure 16.** Bode plot of the loop transfer function, $L(s)$, of the baseline controller, $K_{PI}(s)$ and the SISO plant, $G(s)$ (grey), and the loop transfer of the artificial SISO plant, $\tilde{G}(s)$, in Fig. 14 and $\tilde{K}(s)$ illustrating the effect of the linear combination of both actuators on increasing the bandwidth of the closed-loop system indicated by the vertical lines in the phase plot.



# 4 Results

The FOWT system was simulated in OpenFAST (NREL, 2025a) with the five controllers discussed in Section 3 in environmental conditions of turbulent wind and irregular waves. The simulations were conducted in the above-rated Region 3
($v_{rated} = 11.4$ m/s) at average wind speeds ranging from 12 m/s to 24 m/s, with TurbSim (NREL, 2025b) to generate the turbulent wind field. The irregular waves were generated using JONSWAP spectrum. All the simulations were performed for a simulation time of 1200 s, with the first 600 s neglected for transients.

An example time-domain simulation at a reference wind speed of 18 m/s is illustrated in Figure 17 and Figure 18. The time traces are complemented with the power spectra for a detailed view of the controllers' performance. Looking at the rotor
speed signal in Figure 17, we can see how impressive of a difference the robust tuning of the SISO PI controller can make in comparison to the detuned SISO PI controller. The rotor speed's peak-to-peak amplitude of the Robust SISO is significantly reduced compared to the detuned SISO. This is also evident in the spectral content of its power spectrum, as the rotor speed oscillations are suppressed till 0.1 Hz.

For the SISO controller, the generator torque is kept constant and the generator power in Region 3 is directly related to the
generator speed. The reduction in the rotor speed oscillations reflects on the generator power leading to an improved power quality with less fluctuations. However, such an improved performance comes at the cost of actuation. This is to be expected since the increased bandwidth of the Robust SISO means higher control activity, which can be seen in the blade pitch signal with higher spectral content across the frequency range, leading to an increase in the blade pitch variation.

Regarding the MISO controller in Fig. 17, its main objective is to add damping to the closed-loop system to compensate for
the severe reduction in damping caused by the aerodynamic damping as explained by Eqn. (18) and Eqn. (22). In this work, the MISO controller is composed of the Robust SISO controller, and added to it is the inner feedback loop from the platform pitch rate $\dot{\theta}$ to blade pitch as shown in Fig. 9. The MISO controller in Fig. 17 appears to be doing slightly better than the Robust SISO in a small frequency segment within the low-frequency region before 0.05 Hz, while no significant difference is observed between both controllers at other frequencies. Similar to the detuned and Robust SISO cases, the generator power follows the
same trend as the rotor speed since the generator torque is constant in the case of the MISO controller. This explains the absence of the generator torque curves relevant to the three cases in the power spectrum. The MISO controller blade pitch actuation does not change much from the Robust SISO controller. It simply is a bit more active and thus more oscillatory because of the extra blade pitch input added.

As for the MIMO controller in Fig. 17, the generator torque is employed as an extra actuator to provide parallel compensa-
tion (Skogestad and Postlethwaite, 2005) to the FOWT system to deal with the RHPZs. Implementing the MIMO controller results in a modest enhancement of rotor speed, as the substantial improvement achieved by the Robust SISO controller over the detuned version significantly limits the potential for further error reduction. With the generator torque not constant anymore, the power variation includes contributions from both generator speed and generator torque, showing a clear drawback of the MIMO controller.





**Figure 17.** Non-linear simulation results for the FOWT system, simulated with each of the controllers described in Section 3 at a reference wind speed of 18 m/s.

Transitioning to the newly proposed control structure, the SIMO controller demonstrates superior performance in generator speed regulation—the primary objective of this controller—particularly when compared to the Detuned SISO controller. While one might expect increased blade pitch activity to achieve better generator speed regulation, this is not the case. Instead, the blade pitch action remains nearly identical to that of the Robust SISO, MISO, and MIMO controllers. This is because, beyond a certain point, generator torque takes over, as previously shown in Fig. 15. Consequently, the generator torque response becomes

highly aggressive, exhibiting significant variations to maintain a more stable generator speed signal, even reaching saturation. However, this comes at the expense of power quality, similar to the MIMO controller. Notably, the SIMO controller exhibits an even more aggressive generator torque action than the MIMO controller. A less aggressive tuning of the SIMO controller would



reduce the actuator usage and improve the power quality. Nevertheless, if the power quality is the main control objective, a controller aimed at that objective could be synthesised, but at the cost of increased drivetrain loads (Stockhouse and Pao, 2024).

Across the above-rated wind speed spectrum, the SIMO controller achieves the lowest rotor speed oscillations, as indicated by the standard deviation, without any notable difference in blade pitch action compared to other controllers (see Fig. 19). However, the generator torque experiences a dramatic increase with the SIMO controller, even at wind speeds where the RHPZs are expected to disappear (above 16 m/s). This is because, unlike other controllers, the SIMO controller continuously engages the generator torque actuator across all wind speeds, including those without RHPZs. As a result, variations in generator speed

have a considerable impact on generator power. In the simulations conducted at reference wind speeds of 12–14 m/s, the system occasionally operates below the rated wind speed, leading to fluctuations in generator torque. This occurs despite the Detuned SISO, Robust SISO, and MIMO controllers being designed to maintain a constant generator torque with zero standard deviation in Region 3—a condition that is fully realised at wind speeds above 14 m/s.

Examining Fig. 18 and Fig. 19 simultaneously, it is evident that all controllers reduce platform pitch oscillations compared

to the fluctuations observed with the Detuned SISO. Among them, the MISO controller achieves the greatest reduction, as it is specifically designed to enhance platform pitch damping—an effect clearly visible in the power spectrum around the platform pitch eigenfrequency ($\approx 0.033$ Hz).

Although the SIMO controller is primarily designed to mitigate generator speed fluctuations, it also succeeds in reducing platform pitch oscillations below the Detuned SISO level. While its effectiveness in this regard is lower than that of the MISO

and MIMO controllers, this reduction remains beneficial.

Furthermore, this improvement extends to the tower base fore-aft moment ($M_{TwrBs,y}$), as there is a strong correlation between platform pitch motion and tower base loading. Consequently, controllers that effectively suppress platform oscillations also contribute to significant tower fatigue reduction.

Regarding the blade-root flapwise moment ($M_{Flp,y}$), all controllers outperform the Detuned SISO across all wind speeds,

as shown in Fig. 19. This improvement is evident at low frequencies up to 0.1 Hz, after which there is a slight drop in performance, temporarily exceeding the level of the Detuned SISO. Beyond this point, all controllers converge, exhibiting no significant differences, as depicted in Fig. 18.

Rotor-shaft torsional loading ($\tau_{shaft}$) is a well-known drawback of torque feedback in wind turbine control systems. While both the Robust SISO and MISO controllers exhibit smaller shaft loading excursions compared to the Detuned SISO, the

MIMO and SIMO controllers, which rely on torque feedback, introduce greater fluctuations in shaft torsional loading. As shown in Fig. 18, this effect is particularly pronounced in the SIMO controller, which exhibits elevated shaft loading variations across all wind speeds, as further illustrated in Fig. 19.

Based on these findings, the authors recommend an adaptive approach, where different proposed controllers are alternated depending on environmental conditions and control objectives. For example, at certain times, the turbine operator may priori-

tise minimising generator speed oscillations and activate the corresponding controller. At other times, the focus may shift to reducing structural loading, necessitating a different control strategy. Since no single controller can simultaneously optimise all objectives—some of which may be conflicting—dynamic selection based on operational priorities is advised.



**Figure 18.** Non-linear simulation results for the FOWT system, simulated with each of the controllers described in Section 3 at a reference wind speed of 18 m/s.

Another recommendation is to incorporate a feedforward control strategy to reduce dependence on reactive feedback control. If an accurate preview of disturbances affecting the FOWT is available, a LiDAR feedforward controller (Schlipf et al., 2020) targeting the wind turbulence and a wave feedforward controller (Hegazy et al., 2023b, 2024) targeting the wave forces can be implemented to mitigate the effects of wind and wave disturbances on the FOWT, respectively. This approach alleviates the need for a high-bandwidth feedback controller, as the feedforward controllers would handle most of the disturbance rejection.










**Figure 19.** Controller performance: Non-linear simulation results for the FOWT, simulated with each of the controllers described in Section 3 at a reference wind speed of 18 m/s.

## 5  Conclusion

A new fixed-structure controller has been developed for FOWTs to effectively mitigate the well-known "negative damping" instability and address the non-minimum phase behaviour introduced by the persistent RHPZs in $G_{\Omega\beta}$. Designed specifically for generator speed regulation, the proposed controller was evaluated through non-linear simulations in OpenFAST, where it outperformed the existing FOWT controllers from the literature. Furthermore, it demonstrated robustness in a high-fidelity simulation environment, effectively handling additional system dynamics.





The primary advantage of the proposed FOWT controller is that it operates without requiring any additional sensors, preserving the conventional SISO configuration by relying exclusively on generator speed measurement. This approach enhances robustness, as incorporating extra signals can increase sensitivity to unmodeled dynamics. Additionally, the controller can be regarded as an artificial SISO controller, as shown in Fig. 14 and Fig. 15, where the plant transfer function is pre-filtered to achieve the desired control performance.

While the MIMO controller features a simpler control structure compared to the SIMO controller, the SIMO configuration provides built-in redundancy within the FOWT system, ensuring continued operation in the event of floating platform sensor failure. If the wind turbine is equipped with platform pitch sensors and the MIMO controller is in use, a sensor malfunction could compromise performance. In such a scenario, the SIMO controller acts as a backup solution, allowing the system to operate despite the loss of platform pitch measurements.

Incorporating inner loops into the standard control loop $G_{\Omega\beta}$, whether using MISO, SIMO, or MIMO structures, expands the design space for the SISO PI feedback controller, enabling the achievement of higher bandwidth. However, a well-known drawback of employing generator torque actuation for parallel compensation is the resulting increase in shaft and drivetrain loads (Fischer, 2013), along with deteriorated power quality. To mitigate power quality concerns, alternative MIMO feedback architectures, such as a constant-power controller (Stockhouse and Pao, 2024), can be integrated.

Furthermore, the cost function in the robust control tuning approach from Stockhouse and Pao (2024) has been modified to prevent actuator saturation. Without this adjustment, actuator activity could become unbounded, leading to simulation failure. This refinement has enhanced performance in the primary objectives of generator speed regulation and tower load reduction, even in the presence of modelling inaccuracies resulting from dynamic simplifications and omitted degrees of freedom.

*Code and data availability.* The code and data presented in this work can be made available upon request.

*Author contributions.* AH conceptualisation, methodology, investigation, writing – original daft under the super-vision of PN and JWVW. The insights and conclusion presented in this paper are the results of extensive discussions among the co-authors. All co-authors thoroughly reviewed the article.

*Competing interests.* At least one of the (co-)authors is a member of the editorial board of Wind Energy Science. The authors have no other competing interests to declare.

*Acknowledgements.* This project is part of the FLOATFARM project. The research presented in this paper has received funding from the European Union's Horizon 2020 research and innovation programme under grant agreement no. 101136091.



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
