# Peer review of "Control design for floating wind turbines"

_Wind Energy Science, 2025_

## Referee Comment (RC2)

**Control design for floating wind turbines**

[referee-annotated manuscript omitted]

---

## Author Comment (AC1)

| | |
|---|---|
| Date | September 21, 2025 |
| Contact person | A. Hegazy |
| Telephone | +31 (0)15 27 89201 |
| E-mail | A.R.Hegazy@tudelft.nl |
| Subject | Response to Reviewers |

**Delft University of Technology**

Delft Center for Systems and Control

Anonymous Reviewer #1
Anonymous Reviewer #2
Anonymous Reviewer #3
*Reviewers, Wind Energy Science*

Address
Mekelweg 2
2628 CD Delft
The Netherlands

www.dcsc.tudelft.nl

Dear Reviewers,

We sincerely thank you for your valuable and constructive feedback on our paper. Your comments have greatly assisted us in enhancing the quality of our work. We have carefully considered all of the points raised and revised the paper accordingly. This letter serves to address your comments and provide an overview of the changes made. Below, we will respond to each of your review comments, and at the end of this document, a colour-coded revised version is included showing the changes made to the manuscript.

Yours sincerely,

Amr Hegazy
Peter Naaijen
Jan-Willem van Wingerden

Enclosure(s): Response to comments of Reviewer #1
Response to comments of Reviewer #2
Response to comments of Reviewer #3
Colour-coded revised version

**Response to comments of Reviewer #1**

General comments:

- Referring to your comment about the writing style. This paper was not the result of a work that was written all around the same time. The first half of the paper is much older than as it was written at the beginning of my PhD, while the second part took place at the end of my PhD. All of this was compiled together to get this paper at the end.
- Regarding the gains used, a table with the employed gains was added to the SIMO controller section.

Specific comments:

Reviewer: Abstract: "feedback control": Specify the type, e.g., generator speed or power control.

Authors: *Thank you for noticing this. We have modified the text as follows:*
**Text excerpt:** "generator speed feedback control"

Reviewer: Abstract: "applying the same ... unstable": Clarify that only the pitch-to-feather scheme typically induces instability; other feedback control strategies exist that maintain stability in floating wind turbines.

Authors: *Thank you for your comment. The text was modified as follows:*
**Text excerpt:** "via the pitch controller to feather the blades, is well-established, employing the same controller gains with floating offshore wind turbines causes the turbines to become unstable."

Reviewer: Abstract: "fore-aft motion": Specify this is the nacelle fore-aft motion.

Authors: *Thank you for your comment. The text was modified as follows:*
**Text excerpt:** "the nacelle fore-aft motion."

Reviewer: Abstract: "more robust": Define what is meant by "robustness". Robust to what (e.g., model uncertainties, disturbances)?

Authors: *Thank you for your comment. This is explained in the following sentence in the text.*
**Text excerpt:** "which makes controllers often sensitive to unmodelled dynamics."

Reviewer: Lines 18–19: Remove "this is not . . .  know that" and "no wonder . . .  wind energy". These are unnecessary and informal.

Authors: *Thank you for your comment. The whole paragraph was modified as follows:*
**Text excerpt:** "Wind energy is essential to meeting the decarbonisation objectives of the European Union (EU) energy system, as it ensures delivering clean, affordable and secure electricity to various sectors, including households, industry and transport. Consequently, wind energy is expected to heavily contribute to the EU renewable energy targets, with wind energy already covering $19\%$ of the EU's electricity demand in 2024. This has seen the EU revising the renewable energy directive, which lays down a minimum target of $42.5\%$ share of renewables by 2030 with an aspiration to reach $45\%$. This is $10.5\%$ higher than the initial $32\%$ target. Subsequently, the EU could fulfil its ambition of becoming climate-neutral by 2050 [1]."

Reviewer: Lines 20–26: Consider whether this section is necessary. Is the article relevance limited to the EU context?

Authors: *Thank you for your question. While the article's relevance is indeed not limited to the EU context, the EU was the funding body as mentioned in the acknowledgements. However, a phrase was added to account for other continents as follows:*
**Text excerpt:** "As for the other continents, similar trends are observed [2]."

Reviewer: Lines 37–38: The phrase "while modifications . . .  to reducing the LCOE" is vague. Co-design of turbine and platform may reduce LCOE. Please clarify the mechanism you're referring to.

Authors: *Thank you for your comment. Since it is not clear to the reader, we clarified this by modifying the text as follows:*
**Text excerpt:** "Developing new control architectures can contribute to reducing the LCOE. Another approach is control co-design [3], which has proven to be highly effective."

Reviewer: Line 80 and following: Including physical units is not necessary. Models can use different units.

Authors: *Thank you for your comment. While this is definitely correct, to our understanding, including the units is part of the journal's submission requirements. Furthermore, it helps with reproducing the results of the paper.*

Reviewer: Line 87: "and a non-linear function f(x) = 0". Unclear connection with the surrounding text.

Authors:  *Thank you for your comment. The whole paragraph was modified as follows:*
**Text excerpt:** "with $\rho$ (kg.m$^{-3}$) as the air density, $R$ (m) being the rotor radius, $C_p(\lambda, \beta)$ as the power coefficient, which depends on the blade pitch angle, $\beta$ (rad), and the tip-speed ratio, $\lambda = \Omega R / v N_{gb}$, with $v$ (m.s$^{-1}$) being the wind speed normal to the rotor plane. At steady-state, the non-linear wind turbine dynamics in Eq.(1) can be linearised using first-order Taylor series expansion around an equilibrium point as:"

Reviewer:  Line 98: Instead of "remains constant ...  and assumption errors", state clearly that the control strategy assumes accurate knowledge of $k_g$.

Authors:  *Thank you for your comment. The text was modified as follows:*
**Text excerpt:** "It is fair to mention that the controller in Eq.(4) assumes a constant $k_g$ throughout the wind turbine's lifetime."

Reviewer:  Line 113: "Notice that the terms irrelevant to the control problem". Please specify which terms are considered irrelevant and why.

Authors:  *Thank you for your comment. The text was modified as follows:*
**Text excerpt:** "Equation (3) represents the closed-loop system of the wind turbine in the above-rated region. Therefore, only the terms containing $\psi$ and its derivatives are considered [4]. The remaining terms are irrelevant to the control problem and therefore do not appear in Eq. (6)."

Reviewer:  Line 131: The definition of above-rated operation is incorrect. Rated wind speed is the minimum at which rated power is produced. All higher wind speeds are above-rated.

Authors:  *Thank you for your comment. The definition is actually meant for the steady state thrust in the above-rated region in particular.*

Reviewer:  Line 133: "The steady-state ...  generator speed variations". This appears inaccurate. In above-rated conditions, pitch is increased to limit aerodynamic torque. Clarify.

Authors:  *Thank you for your comment. The text was modified as follows:*
**Text excerpt:** "The steady-state blade pitch angle varies along the operating curve to limit the aerodynamic torque and reduce the oscillation in the power production. The generator torque is kept constant instead of constant power since this strategy limits the generator speed variations, hence, reduces drive train loads and pitch activity [5]."

Reviewer:  Line 135: "The objective is to achieve ...  differential diminishes". Unclear. Please rephrase for clarity.

Authors:   *Thank you for your comment. The sentence was removed since it did not fit in this paragraph.*

Reviewer:   Line 138: "capturing the critical dynamics". Specify what these critical dynamics are. Are you referring to adding platform pitch to drivetrain dynamics? State clearly which degrees of freedom are included and which are not, along with your rationale.

Authors:   *Thank you for your comment. The text has been modified as follows:*
**Text excerpt:** "To form a FOWT mathematical model, the generic 1-DOF model of the wind turbine in Eq. (3) is combined with the floating platform dynamics. For the sake of explaining the negative damping problem analytically, only a 2-DOF FOWT model capturing the critical dynamics is used, where the platform pitch degree of freedom (DOF) is primarily considered to characterise platform dynamics, as the negative damping instability is most pronounced at the platform pitch eigenfrequency, where there is no damping from the mooring and vary little hydrodynamic damping leading to negative damping if the pitch control is fast [5, 6]."

Reviewer:   Line 149: "omitting radiation damping ... frequency-dependent coefficients". This choice is questionable. Why are some terms relevant only in certain scenarios? Clarify.

Authors:   *Thank you for your comment. For a spar platform, the radiation damping is relatively small compared to the viscous damping. It is almost equal to zero across all frequencies except around the natural frequency of the mode of interest. Regarding the added mass, it can be assumed constant across all frequencies, which is pretty much the case. Therefore, computing the added mass using Morison's equation is valid. For a semi-sub (triple spar), the same assumption applies, whereas for other platforms with different geometries, it might not be the case. The references .[7, 8] explain the reason for the omission.*

Reviewer:   Line 181: "clearly shows ... via the term". Why is this term important?

Authors:   *Thank you for your comment. The whole sentence was removed as its importance emerges later in section 3.3 where it is discussed.*

Reviewer:   Line 192: "is of the main interest". Justify why this is the main interest.

Authors:   *Thank you for your comment. The text was modified as follows:*
**Text excerpt:** "For the feedback control of FOWTs in Region 3, the control objective is to reduce the generator speed oscillations using the blade pitch action. Consequently, the TF, $G_{\Omega,\beta}$, mapping the blade pitch angle to the generator speed, in Eq. (16) is of the main interest:"

Reviewer: Line 205: "quasi-static ... wind speed". Define quasi-static equilibrium more precisely. What does it mean that system variables are "balanced"?

Authors: *Thank you for your comment. The text was modified as follows:*
**Text excerpt:** "In a closed-loop FOWT system at steady state, the gradient $dF_a/dv$ is positive below-rated wind speed, meaning the thrust force increases as wind speed rises."

Reviewer: Line 208: "modifies the force direction". Unclear. What force? How is the direction modified?

Authors: *Thank you for your comment. The part was removed as it was incorrect.*

Reviewer: Line 229: "the pitch angle ... are negative". I think you are mixing gradients and steady state values. The derivatives of thrust and aerodynamic torque w.r.t. blade pitch are negative. As the wind speed increases, the blade pitch is drecreased to have constant aerodynamic torque (hence to balance the increase it would have because of wind speed). The increase in blade pitch doesn't yield constant thrust but thrust decreases.

Authors: *Thank you for your comment. We agree with the thrust and aerodynamic gradients being negative wrt blade pitch. However, we beg to differ with the rest of the sentence, as when the wind speed increases, the blade pitch increases too to keep the balance between the aerodynamic and generator torques, and avoid accelerating the rotor. The aerodynamic torque decreases too in the process.*

Reviewer: Line 293: "this approach". Please clarify which approach is being referred to.

Authors: *Thank you for your comment. The text was modified as follows:*
**Text excerpt:** "Applying the global detuning approach."

Reviewer: Line 299: The explanation regarding gain tuning and stability is not clear. Rephrase.

Authors: *Thank you for your comment. The text was modified as follows:*
**Text excerpt:** "Rather than applying a global detuning strategy at all the operating points as described in the previous section, a more effective method involves individually tuning the PI controller for the fastest achievable response at each operating point, while still ensuring the stability of the linear system [9, 10, 11, 12]. However, achieving a stable system is not sufficient in control design; the system must also exhibit adequate stability margins, which indicate how close it is to instability and how robust it is to disturbances."

Reviewer: Line 313: "a range of ... $\zeta_c$". Define these parameters explicitly.

Authors: *Thank you for your comment. The text was modified as follows:*
**Text excerpt:** "a range of the Proportional-Integral (PI) control parameters, namely, the natural frequency $(\omega_c)$ and the damping ratio $(\zeta_c)$"

Reviewer: Line 321: The effects of $\omega_c$ and $\zeta_c$ on bandwidth and stability are difficult to interpret without definitions. Please clarify.

Authors: *Thank you for your comment. They were defined as per your comment above. They were also introduced and defined in section 2 in the equations from 6 to 8.*

Reviewer: Line 326: What is $\omega_p$ and how does it differ from $\omega_c$? Define all parameters clearly.

Authors: *Thank you for pointing this one out. There was a typo and it should have been $\omega_c$.*

Reviewer: Line 336: "regularisation terms". Briefly explain what regularization is used for in this context.

Authors: *Thank you for your comment. The text was modified as follows:*
**Text excerpt:** "Regularisation terms may be added to the objective function to fulfil control objectives such as minimising the generator speed and power oscillations, as well as reducing the loads [9], and limiting the control gains [12]."

Reviewer: Lines 371, 377: "blade pitch damping" ???

Authors: *Thank you for your comment. The title was changed to "additional blade pitch loop".*

Reviewer: Line 400: "should this objective ... it will be unstable". Unclear. Explain what the objective is and why instability would result.

Authors: *Thank you for your comment. The text was modified as follows:".*
**Text excerpt:** "Should the objective of decoupling the drivetrain and the platform dynamics be sought, extra filtering is required to change its dynamics; otherwise, the system damping worsens and it becomes unstable."

Reviewer: Line 463: "It was learnt ... band-pass filter". Why is it feasible to retain the PI controller and what is the role of the band-pass filter?

Authors: *Thank you for your comment. As mentioned in the text, those control structures for the blade pitch and the generator torque, were obtained based on*

$$\mathcal{H}_\infty$$

*optimisation as explained in [13]. First, the structure of the different control loops was learnt via $\mathcal{H}_\infty$ synthesis. For the blade pitch the returened structure was more of a low-pass filter, while a band-pass filter for the generator torque. Afterwards, the structure of both was fixed to basic control elements, ensuring the same performance. Therefore, the blade pitch loop retained the PI structure, and the generator torque became an inverted-notch filter. We strongly, advise you to refer to this publication [13]. The text was also modified as follows:*

**Text excerpt:** "It was learnt from $\mathcal{H}_\infty$ control synthesis that the blade pitch maintains the PI structure, while the generator torque requires an inverted-notch filter hegazy2023. From its name, an inverted-notch filter is a control element that is only operational at a specific frequency. Thinking about it, such a control structure, resulting from the $\mathcal{H}_\infty$ synthesis, for the generator torque control loop is reasonable since the generator torque input should not operate across all frequencies, but only around the RHPZs frequency where the blade pitch input is blocked. Therefore, the generator torque control loop takes over."

Reviewer: Line 468: "an inverted notch". Clarify the purpose of using an inverted notch filter.

Authors: *Thank you for your comment. This was explained in the previous comment and the text was modified accordingly.*

Reviewer: Line 519: "to compensate for . . . aerodynamic damping". Rephrase for clarity. What is being compensated, and how?

Authors: *Thank you for your comment. The text was modified as follows:*

**Text excerpt:** "its main objective is to add damping to the closed-loop system, through extra blade pitch action, to compensate for the severe reduction in the overall system damping caused by the negative aerodynamic damping"

Technical corrections:

Reviewer: Line 42: Remove "peak to peak".

Authors: *This part has been removed as advised.*

Reviewer: Line 80: Remove "(kg.m2)" and similar notations. The dot is not conventionally used.

Authors: *The dot was removed from the units as advised.*

Reviewer: Line 121: Replace "lie in the vicinity of" with "include".

Authors: *The text was modified as advised.*

Reviewer: Line 272: Remove "a point that is elaborated further in this section". It's redundant.

Authors: *The text was removed as advised.*

Reviewer: Line 277: Replace "triggered by" with "associated with"; replace "causing" with "causes".

Authors: *The text was modified as advised.*

Reviewer: Line 320: Consider replacing "failure" with "instability" if that's what you meant.

Authors: *The text was modified as advised.*

Reviewer: Line 325: Replace ":" with ",".

Authors: *The text was modified as advised.*

Reviewer: Lines 367–368: Replace "actuators" with "actuation".

Authors: *The text was modified as advised.*

Reviewer: Figure 15: Adjust the line style for  to improve readability.

Authors: *Line style has been adjusted as advised.*

Reviewer: Figure 15 caption: Explain which transfer function the Bode plot represents.

Authors: *Caption has been modified for further explanation as follows:*
**Text excerpt:** "Bode plot of the normalised MISO plant $\bar{G}$ illustrating the frequency response of each control channel separately (solid lines) as well as the response of the SISO plant $\tilde{G}$ in case of the linear combination of both actuators (dashed line) where the blade pitch actuator is active till a certain frequency before its authority deteriorates, thus, the generator torque actuator takes over from that frequency onwards."

Reviewer: Figure 16, magnitude: I think one dot is missing. Figure 16, phase: one vertical line is hard to see.

Authors:   *Thank you for your comment. At first glance, it seems that the dot corresponding to the gain margin is missing for the artificial SISO plant ($\hat{G}$). However, it is not the case since for this linear transfer function, at a certain operating point, and after the modifications that have been applied, the gain margin is infinite. Regarding the phase, the vertical line illustrating the phase margin is hard to see since it is actually very small, which indicates less controller robustness.*

Reviewer:   Line 510: Replace "how impressive of a difference" with "the significant impact".

Authors:   *The text was modified as advised.*

Reviewer:   Line 531: Remove repeated citation. Only cite once unless needed.

Authors:   *Repeated citations were removed as advised.*

Reviewer:   Figure 17: Reduce line thickness to avoid covering differences.

Authors:   *Those lines were removed as per your suggestion and considered before Table 3.*

Reviewer:   Line 547: Replace "dramatic" with "large".

Authors:   *The text was modified as advised.*

**Response to comments of Reviewer #2**

General comments

Reviewer:   Clarify the turbine model used (which platform, rating, etc.).

Authors:   *Thank you for noticing this. The text was modified to include this important piece of information in section 2.1 and again in the results section.*

Reviewer:   Clarify the site conditions assumed, i.e. wave height, period, wind shear, TI.

Authors:   *Thank you for your comment and advice. The text was modified to clarify this in the results section.*

Reviewer:   Clarify the controller architecture and changes of existing controllers from the literature.

Authors:   *Thank you for your comment. All the control architectures are explained analytically and graphically, as illustrated by the figures and equations corresponding to each controller.*

Reviewer:  Tabulate used controller gains for reproducibility of results.

Authors:  *Thank you for your comment and advice. A table with all the gains was added as advised.*

Reviewer:  Clarify structure of new SIMO controller, especially why the plant is changed. I expected only K to change, not the plant.

Authors:  *Thank you for your comment. The modification applied to the plant primarily aimed to bring the loop gain of both input channels to unity, ensuring they have comparable gains and that their optimisation-based tuning could be properly performed.*

Reviewer:  Discuss sensitivity to sea state: Often controllers perform well in benign sea states but fail in harsher conditions. It should be proven that the controllers don't amplify first-order wave motions.

Authors:  *Thank you for your comment. The sea state was defined in the results section as previously advised, where a challenging sea state was used in the simulations. Apart from that, the controller is actually blind to the first-order wave effects as they do not appear in the response since they operate at higher frequencies than the controller's natural frequency, while the low-frequency dynamics are of interest. We refer you to [14] where this subject was explained through wave tank experiments.*

Reviewer:  Improve comparability of the results: Some controllers use a lot of generator torque actuation, while others don't.

Authors:  *Thank you for your comment. This has to do with the tuning and also to showcase that the SIMO controller can better reduce rotor speed oscillations at the expense of more generator torque actuation.*

**Specific comments**

Reviewer:  Line 38: "should not be overlooked as it can significantly contribute to reducing the LCOE." How?

Authors:  *Thank you for your question. The text was modified as follows:*
**Text excerpt:** "Developing new control architectures can contribute to reducing the LCOE. Another approach is control co-design [3], which has proven to be highly effective."

Reviewer:   Line 56: "control channels are separately tuned to achieve improved dynamic responses of a specific output." Clarify what this means exactly. What is a "control channel"? Is compartmentalised the same as "segmented" control? How does it relate to the "multi-SISO" approach (see Fleming2016) and the terms "sequential", or "decentralised" control?

Authors:   *Thank you for your comment. The text was modified for clarification as follows:*
*Text excerpt:"Systems with more than one actuating control input and more than one sensor output may be considered as multivariable systems or Multi-Input-Multi-Output (MIMO) systems. The control objective for multivariable systems is to obtain a desirable behaviour of several output variables by simultaneously manipulating several input channels. A FOWT is a MIMO system. To evaluate such a system, MIMO transfer function matrix is needed. In many FOWT control strategies, the feedback control loops are often designed separately in a decoupled format, with each control loop tuned to improve the response of a specific output [15]. This means that the multivariable controller design is reduced to a series of single-loop controllers. Although this approach is common, the loops in a MIMO system are dynamically coupled. As a result, changing the settings of one control loop can affect the behaviour of other loops, causing interaction between them. Subsequently, the interaction between the different control loops of a system should be dealt with simultaneously."*

Reviewer:   Line 65: tutorial or a review?

Authors:   *Thank you for your question. It was changed to "review" instead.*

Reviewer:   Line 105: introduce $\Omega_{rat}$.

Authors:   *Thank you for noticing this. The text was modified as follows:*
*Text excerpt:"At above-rated wind speeds (referred to as Region 3), a conventional wind turbine controller relies on the blade pitch to regulate the generator speed to its rated value, $\Omega_{rat}$."*

Reviewer:   Equation 6: Clarify if the generator torque is constant.

Authors:   *Thank you for your comment. This is mentioned in the paragraph above Eq. 6.*

Reviewer:   Line 113: does this mean that changes in relative wind speed are irrelevant? How can you then model negative damping?

Authors:   *Thank you for your comment. The changes in the relative wind speed are included in Eq. 13 due to the rigid body motion of the floater.*

Reviewer:   Line 150: clarify which types of models you're using. What is the "control model", as opposed to the "simplified 2D model"

Authors: *Thank you for your comment. The text was modified as follows:*
**Text excerpt:**"However, to preserve key dynamic couplings, the control model used for the control design must include additional modes that capture the most significant system dynamics, namely the platform's surge and heave, and the tower first fore-aft bending [9]; otherwise, some interactions within the system may be overlooked [10]."

Reviewer: Equation 11: How about surge motion?

Authors: *Thank you for your comment. The surge motion is not as problematic as the pitch since the negative damping instability arises due to the pitch motion. The system zeros corresponding to the surge motion always lie in the left half plane. This we already showed in [13]. So for the sake of the analytical model, we excluded the surge motion in the simplified 2D model. However, in the control model used for the synthesis, the surge motion was included.*

Reviewer: Line 206: "gradient $dF_a/dv$ is positive below-rated wind speed" Clarify if you're referring to an OL or CL system.

Authors: *Thank you for you comment. The text was modified as follows:*
**Text excerpt:**"In a closed loop FOWT system at steady state, the gradient dFa/dv is positive below-rated wind speed" Clarify if you're referring to an OL or CL system."

Reviewer: Line 209: "As a consequence, aerodynamic damping is positive at below-rated wind speeds but turns negative at above-rated wind speeds." Note that this is only true in CL.

Authors: *Thank you for your comment. This was clarified in the previous comment.*

Reviewer: Equation 22: Why is there the torque in the equation of the thrust?

Authors: *Thank you for your comment. From Eq. 19, where $\Omega_r$ is constant, an expression was obtained for the blade pitch differential in Eq. 20. Plugging Eq. 20 into Eq. 21, reach such an expression in Eq. 22, which is also derived in Eq. 18.*

Reviewer: Line 249: "A zero represents a critical frequency, referred to as the frequency of the zero" This sentence is unclear.

Authors: *Thank you for noticing this. The text was modified as follows:*
**Text excerpt:**"The roots of the numerator of a transfer function are called zeros (denoted by ◯ in Fig. 3)."

Reviewer: Line 293: "bandwidth" The bandwidth is only constant if the platform pitch natural frequency does not change across operating points.

Authors: *Thank you for your comment. The platform pitch natural frequency is a system property and does not change across the operating points. The detuning process initially was applied such that the bandwidth stays constant [16, 17] across all the operating points, such that its value does not exceed the platform pitch natural frequency.*

Reviewer: Line 354: "The optimisation goal is to minimise" repetition.

Authors: *This part was removed as advised.*

Reviewer: Line 46: Doesn't this mean that rotor speed changes resulting from freestream wind have the same effect?.

Authors: *Thank you for your question. The main point of developing this controller is to prove that the parallel compensation can still be applied without the platform pitch motion sensor. The rotor speed response to the wind disturbance changes since the controller has a different structure from the other controllers presented before, which can be seen in the results section. If we are looking at the frequency response from the wind disturbance to the rotor speed in a Bode plot, there will be no change since the controller was synthesised to have generator speed error as input and a combination of blade pitch and generator torque output. However, the relative wind speed effect, due to the platform pitch motion in particular, is the one that is going into the controller, and being reduced.*

Reviewer: Line 468: Above, you say that the platform pitch measurement is not needed, which is the measurement going into this controller?

Authors: *Thank you for your question. Indeed for the SIMO controller, the platform pitch measurement is not needed, as it only rely on the generator speed measurement as illustrated in Fig. 13.*

Reviewer: Figure 14: What does $\bar{G}(s)$ stand for?.

Authors: *Thank you for noticing this one. It has been defined in the text now. It stands for the normalised plant. The text was modified as follows:*
**Text excerpt:**"In order to do that, the original MISO plant $G(s)$ is normalised to $\bar{G}$ such that the magnitude of both the blade pitch and the generator torque input channels becomes unity so that both control inputs are of comparable effect."

Reviewer: Line 504: Which is the FOWT design used, rated power, what is the wave height?.

Authors:  *Thank you for noticing that. The FOWT design used has been introduced earlier in the text and added also here together with the environmental conditions. The text was modified as follows:*
**Text excerpt:**" The FOWT system (NREL 5-MW RWT jonkman2009definition atop OC3 floater jonkman2010definition) was simulated in OpenFAST Openfast with the five controllers discussed in Section ?? in environmental conditions of turbulent wind and irregular waves. The simulations were conducted in the above-rated Region 3 ($v_{rated} = 11.4$ m/s) at average wind speeds ranging from 12 m/s to 24 m/s, with TurbSim TurbSim to simulate the turbulent wind field, where the International Electrotechnical Commission (IEC) Kaimal spectral model was used as a turbulence model with a turbulence intensity of $14\%$. The irregular waves were generated using JONSWAP spectrum at a significant wave height $H_s = 3$ m and peak period $T_p = 12$ s."

Reviewer:  Figure 17: The first-order wave response is not visible here. Are waves enabled in the simulation?

Authors:  *Thank you for your question. The first-order wave effects do not appear in the response as they operate at higher frequencies than the controller's natural frequency, and thus the controller do not react to the incoming waves although they are enabled in the simulation.*

Reviewer:  Line 605: Why does the SIMO controller saturate, but the others don't (i.e. they use smaller gains which prevent saturation). Wouldn't it make sense to use the same strategy for all?.

Authors:  *Thank you for your question. If we look at the relative gain array of plant at zero frequency for above rated wind speeds, or in other words compare the loop gain of the transfer function mapping the generator torque to the generator speed ($G_{\Omega,\tau_g}$) to that of the one mapping the blade pitch to the generator speed ($G_{\Omega,\beta}$), we will find that the loop gain of $G_{\Omega,\beta}$ is significantly larger than the one of $G_{\Omega,\tau_g}$ indicating that $G_{\Omega,\tau_g}$ requires much higher gains at above rated conditions to be able to do something. Maybe a some reduction in the gain might prevent saturation, but that indeed this is the challenge facing this controller.*

**Response to comments of Reviewer #3**

When we first started writing this paper, it was not the plan to include the sections before section 3.3.3 (the key section, as you call it). However, as the writing progressed, we thought of adding sections 3.1, 3.3.1 and 3.3.2. But we thought that it would be nice to compare the new controller against the existing controllers out there addressing the same control problem. Whereas, section 3.2 was a late addition, and we thought that it would make the comparison a bit more complete, although it is not the case since there are a few still missing, mainly from the CSM MIMO paper as per your nomenclature, but we decided to stop at that point. Having decided to add those additional sections to draw a more complete picture, this is how this paper turned out to be.

Reviewer:  The authors of WES-2025-68 refer to their paper as a tutorial, e.g. on page 3, line 65, they state "This paper provides a tutorial on the design of closed-loop controllers for FOWTs . . ." and indeed they go into some detail in reviewing the various controllers. Given that they cite a different paper in the same special IEEE CSM issue, it is surprising that they don't cite the main tutorial paper in this CSM special issue. It would be useful for the authors of WES-2025-68 to explain why they believe another (very similar) tutorial paper is needed so soon after the 30-page October 2024 tutorial paper in the IEEE CSM.

Authors:  *Thanks for your comment. When developing new controllers, there must be references to compare the performance of the new controllers against. That is why we included those previously proposed controllers. Having said that, we cite the ACC paper and this paper has considerable similarities with the CSM tutorial paper*

Reviewer:  Figure 1 of WES-2025-68 looks quite similar (even in style) with Figure 3 of the CSM MIMO paper. Of course, I do expect that authors may have similar wind turbine diagrams; and I only noted this particular similarity after having marked many of the other similarities enumerated below.

Authors:  *Thank you for your comment. Looking at Figure 3 of the CSM MIMO paper in depth, we are 100% that different tools were used to generate both figures. They might seem similar at first glance, but the similarity is not clear.*

Reviewer:  Figure 3 of WES-2025-68 is quite similar to Figure 2 of the main CSM tutorial paper. The numerical values are not exactly the same, as the floating wind turbines considered are different in the two papers.

Authors:    *Thanks for you comment. We have the same figure in our IFAC paper, which was published before the CSM tutorial paper, and in the field, it is really common to use those kinds of figures. The difference in this paper is that the system has a lower number of states, and we show only the upper half of the figure. Anyway, we have modified the figure to show the upper and lower half planes.*

Reviewer:    The particular architectures reviewed in WES-2025-68, as illustrated in its Figures 4, 9, and 11, are special cases of Figure 2 in the CSM MIMO paper and are also separately discussed in the CSM MIMO paper. A summary discussion is also provided in the main CSM tutorial paper (e.g. Figure S12).

Authors:    *Thanks for your comment. The particular architectures in Figures 4, 9 and 11 were in our IFAC paper, which was published before the two papers being referred to.*

Reviewer:    Lines 54 to 59 of Page 2 in WES-2025-68 read: "A multi-loop feedback system evaluation requires a Multi-Input, Multi-Output (MIMO) transfer function representation. Those multi-loop FOWT control strategies in the literature often employ a compartmentalised feedback design, where individual control channels are separately tuned to achieve improved dynamic responses of a specific output. While this segmented tuning methodology remains widespread, inter-loop dynamic coupling inherent in MIMO architectures generates cross-channel interference phenomena, whereby localised parameter adjustment in a single control loop perturbs the closed-loop response characteristics of adjacent feedback channels." A similar paragraph in the right-hand column on page 64 (in the special issue) of the previously published 2024 CSM MIMO paper reads: "Many multiloop FOWT control design approaches in the literature use single-loop closure, where each loop is designed in isolation with simplifying assumptions, typically to target improvement of a single-output behavior. For example, the common constant-power control loop is designed to decrease the dependence of generator power variations on generator speed variations, but it also has the side effect of reducing the natural stability of the FOWT system. Tuning each loop in isolation is common practice, although coupling between multiple loops means that tuning of one loop most often induces changes in the behavior targeted by another, and a tradeoff is necessary to satisfy multiple performance objectives."

Authors:    *Thank you for your comment. Both paragraphs give pretty much the same meaning, but they are not similar. Moreover, we already cited the CSM MIMO at the end of the paragraph. Furthermore, this is common knowledge that can be found in any textbook about multivariable control. Anyway, the paragraph was rephrased as follows:*

**Text excerpt:**"Systems with more than one actuating control input and more than one sensor output may be considered as multivariable systems or Multi-Input-Multi-Output (MIMO) systems. The control objective for multivariable systems is to obtain a desirable behaviour of several output variables by simultaneously manipulating several input channels. A FOWT is a MIMO system. To evaluate such a system, MIMO transfer function matrix is needed. In many FOWT control strategies, the feedback control loops are often designed separately in a decoupled format, with each control loop tuned to improve the response of a specific output. This means that the multivariable controller design is reduced to a series of single-loop controllers. Although this approach is common, the loops in a MIMO system are dynamically coupled. As a result, changing the settings of one control loop can affect the behaviour of other loops, causing interaction between them. Subsequently, the interaction between the different control loops of a system should be dealt with simultaneously."

Reviewer:  Equation (14) of WES-2025-68 is the same as Equation (14) in the main CSM tutorial paper, except the orders of the states and inputs have been changed. Similarly, Table 1 in WES-2025-68 is the same as Table 1 in the main CSM tutorial paper, except with some different symbols and notation used. While control-oriented models may end up being of similar forms, it is quite uncanny how the presentation of the model in WES-2025-68 is also so similar to that in the previously published main CSM tutorial paper.

Authors:  *Thank you for your comment. Table 1 was removed, and the model presentation is based on our IFAC paper except that we excluded the platform surge DOF.*

Reviewer:  The discussion and analysis of the condition for non-minimum phase zeros on pages 9-12 of WES-2025-68 are similar to both the main CSM tutorial paper (pages 40-42 of the CSM special issue) and the CSM MIMO paper (page 67, which references the main CSM tutorial paper). All of these analyses are based on previous work, especially that of [18].

Authors:  *Thank you for your comment. This discussion and analysis is also similar to our IFAC paper from 2023. The analyses are indeed based on the work of [18], and we stated that explicitly.*

Reviewer: The stability margin discussion on pages 14-16 of WES-2025-68 is very similar to the Robust Controller Tuning section in the CSM MIMO paper (page 69). In particular, the contour plots on page 15 of WES-2025-68 are very similar to those in Figures S1 and S2 in the CSM MIMO paper, again with different specific numerical results because different floating wind turbines are considered in these papers. The authors of WES-2025-68 have added an additional term in their cost function (Equation (24)) to directly address control effort (control saturation), while the authors of the CSM MIMO paper discuss "constraining or regularizing the controller gains in the tuning optimization" to mitigate controller saturation (page 75).

Authors: *Thank you for your comment. Somewhere in the text, a referral to the CSM MIMO paper was added, explaining that this specific section is built upon it. Consequently, the figures on page 15 are similar, while the one on page 16 is new, corresponding to the modification of the objective function. Regarding the regularisation, that was already acknowledged in the text, mentioning that the process was not clearly explained in the CSM MIMO paper when we tried to reproduce the results. That led to the modification of the cost function to constrain the controller gains explicitly.*

Reviewer: Section 3.3.1 in WES-2025-68 is very similar to the Blade-Pitch Platform Damping section of the CSM MIMO paper (pages 73-74 in the CSM special issue). Given the overlaps that I was noticing, I also looked up the Stockhouse et al., Wind Energy, 2024 paper4 (which I will refer to as the WE paper) and note that this Section 3.3.1 of WES-2025-68 has similar ideas as Section 3.1.2 in the WE paper. In particular, the idea of tuning the "strength" of $k_\beta$ via a $\xi_\beta$ parameter that varies from 0 to 1 already appeared in the WE paper with the $\alpha_\beta$ parameter in Equation (18) in the WE paper. While perhaps the authors of WES-2025-68 are meaning to review this MISO control structure from past papers, they do not cite either the CSM MIMO paper or the WE paper in Section 3.3.1 in WES-2025-68. They do cite (Jonkman, 2008; van der Veen et al., 2012) early in Section 3.3.1, but the tunable "strength" of $k_\beta$ does not appear in either of these much earlier papers.

Authors: *Thank you for your comment. The idea of tuning the strength $k_\beta$ via $\xi_\beta$ also appeared in our IFAC paper in 2023. Anyway, a citation was added for the tunable strength.*

Reviewer: Similarly, Section 3.3.2 in WES-2025-68 is very similar to the Parallel Compensation section of the CSM MIMO paper (pages 74-75), including the tunable $\xi_{\tau_g}$ parameter (which is $\alpha_{comp}$ in the CSM MIMO paper and $\alpha_\tau$ in the WE paper).

Authors:  *Same as above.*

Reviewer:  Lines 573-577 on page 30 of WES-2025-68 state: "Based on these find-
ings, the authors recommend an adaptive approach, where different proposed
controllers are alternated depending on environmental conditions and control
objectives. For example, at certain times, the turbine operator may priori-
tise minimising generator speed oscillations and activate the corresponding
controller. At other times, the focus may shift to reducing structural loading,
necessitating a different control strategy. Since no single controller can simul-
taneously optimise all objectives—some of which may be conflicting—dynamic
selection based on operational priorities is advised."
The conclusion of the CSM MIMO paper (page 79) states: "Four multi-
loop FOWT-control approaches have been analyzed and compared to a sta-
ble single-loop baseline controller. The simulation performance of these con-
trollers shows the tradeoffs in designing a multiloop controller. Wind energy
producers must balance instantaneous power-regulation demanded by grid
operators while ensuring operational safety and component longevity for the
lifetime of a wind farm. Multiloop control designs can schedule the usage
and combination of multiple control loops at different points of the operating
region to garner performance benefits, while mitigating drawbacks of each
control strategy. The control approaches taken in this work are intended to
serve as a basis for the intuitive understanding of the impact of structured
multiloop control on FOWT system dynamics."

Authors:  *Thank you for your comment. For a paper comparing different controllers,
and for someone who has an interest in the control of floating wind turbines,
you would know that you can not achieve everything with one single con-
troller. Therefore, such a conclusion is normal to arrive at.*

Reviewer:  Section 3.3.3 may be the key section that appears to be new in WES-2025-68,
and is based on the authors' earlier 2023 IFAC World Congress paper. So a
question is whether such a full-length paper is warranted for the contribution
in Section 3.3.3, and perhaps the addition of a control effort term to the
objective function used in the optimization.

Authors:  *Thank you for your question. Based on the previous comments, it seems that
the only part considered in the IFAC paper is section 3.3.3, although there are
other sections, which are simply sections 3.1, 3.3.1, 3.3.2 that were totally
neglected. Moreover, it is to our understanding that using your own previ-
ously published work is not against the journal's policy as long as it is built
upon and extended, which we simply did here. Additionally, we need the other
controllers to compare against, which was also done in the CSM MIMO paper.*

Reviewer:  While I do realize that accessing the IEEE Control Systems Magazine requires a subscription or membership in the IEEE Control Systems Society, whereas the Wind Energy Science journal is an open-access journal, many major universities with multiple engineering departments have institutional subscriptions to IEEE Xplore that include the IEEE Control Systems Magazine. Further, many control systems researchers are indeed members of the IEEE Control Systems Society. A question is then: what are the rules for overlapping material and ideas with already-published papers, possibly in journals that require a subscription? Do the above similarities give others pause, or am I an outlier in finding this amount of overlap disturbing? My own opinion is that newly submitted manuscripts should not have substantial overlap with previously published papers, regardless of whether they are open-access or not.

Authors:  *Thank you for this question. The two papers that you referred to were based on the papers published at the 2023 ACC as you mentioned below, which is not against the rules of IEEE. We also have had our paper published in 2023 in IFAC, and we are extending it while sticking to the rules of Wind Energy Science.*

Reviewer:  I spent quite some time on this review, but really wanted to be as careful as possible in forming this opinion before writing all of this up. I found that the October 2024 IEEE CSM special issue is largely based on a special so-called "tutorial session" at the 2023 American Control Conference (ACC), and earlier versions of both the CSM MIMO and main CSM tutorial papers were published at the 2023 ACC (and also available via IEEE Xplore), so many of these ideas were already published in 2023.

Authors:  *Thank you for your comment. Our IFAC paper was also published at the 2023 IFAC [13].*

60    output may be considered as multivariable systems or Multi-Input-Multi-Output (MIMO) systems. The control objective for multivariable systems is to obtain a desirable behaviour of several output variables by simultaneously manipulating several

input channels. A FOWT is a MIMO system. To evaluate such a system, MIMO transfer function matrix is needed. In many FOWT control strategies, the feedback control loops are often designed separately in a decoupled format, with each control loop tuned to improve the response of a specific output.  (Fleming et al., 2012). This means that the multivariable controller design is reduced to a series of single-loop controllers. Although this approach is common, the loops in a MIMO system are dynamically coupled. As a result, changing the settings of one control loop can affect the behaviour of other loops, causing interaction between them. Subsequently, the interaction between the different control loops of a system should be dealt with simultaneously. It was demonstrated that improved performance could be achieved when optimally tuning all the control loops collectively, accounting for the interdependencies within the MIMO feedback structure rather than tuning each control loop independently (Stockhouse et al., 2024b). Modern multivariable control methodologies employing state-feedback architectures, including Linear Quadratic Regulator (LQR) (Namik et al., 2008) and $\mathcal{H}_\infty$ control (De Corcuera et al., 2012; Hegazy et al., 2023a) demonstrate systematic efficacy in achieving specified stability and performance envelopes for complex dynamical systems (Skogestad and Postlethwaite, 2005).

This paper addresses the design of closed-loop controllers for FOWTs . We begin by outlining the fundamental principles of closed-loop control for FOWTs and describing how the negative damping problem arises. This instability has been widely recognised as a critical challenge, as it can compromise both performance and structural safety.

To provide context, we review existing control strategies proposed in the literature and assess their ability to mitigate the negative damping effect. Although these approaches have advanced the understanding of the problem, they often rely on additional sensors.

The main contribution of this paper is the introduction of a novel controller structure  that eliminates the need for  supplementary sensors while maintaining robust and reliable performance. A detailed tuning methodology is also presented to support practical deployment. Together, these contributions highlight a new pathway for addressing the negative damping problem in FOWTs and advancing the development of effective control solutions.

The remainder of this paper is structured as follows: In Section 2, the FOWT control problem is defined, and the control design model is introduced. In Section 3, conventional Single-Input, Single-Output (SISO) and MIMO control strategies are discussed. In Section 4, the controllers are evaluated by simulating the closed-loop system using the non-linear aero-servo-hydro-elastic tool OpenFAST (NREL, 2025a).

[revised manuscript text omitted]

The roots of the numerator of a transfer function are called zeros (denoted by $\circ$ in Fig. 3). A zero represents a critical frequency, referred to as the frequency of the zero, where the input signal is entirely blocked and has no effect on the system's output. In particular, RHPZs exhibit an "inverse-response behaviour," meaning the system output initially moves in the opposite direction

[Figure]

**Figure 4.** Block diagram of the FOWT closed loop system, where $G(s)$ represents the plant model, and $K_{PI}(s)$ represents the collective blade pitch controller.

of the expected response (Skogestad and Postlethwaite, 2005). This unique characteristic imposes strict constraints on control system design, especially in single-input single-output (SISO) configurations (Lemmer et al., 2016). Additionally, when the system is excited at or near the frequency of the zero, the risk of instability increases significantly. To mitigate this, limiting the controller bandwidth below the smallest RHPZ frequency is a must (Skogestad and Postlethwaite, 2005).

[revised manuscript text omitted]

the original MISO plant $G(s)$ is normalised to $\bar{G}$ such that the magnitude of both the blade pitch and the generator torque input channels becomes unity so that both control inputs are of comparable effect. Afterwards, a linear combination of the two

525 control elements $\bar{K}_{\tau_g}$ and $\bar{K}_\beta$ is combined with normalised MISO plant $\bar{G}$. This is depicted in Fig. 14 where the extra blocks are integrated with the plant such that there is a new plant  $\tilde{G}(s)$. Therefore, the new SISO plant  $\tilde{G}(s)$ is the result of the linear combination of both control channels as:

$$\underline{G^*}\tilde{G}(s) = \underline{G}\bar{G}(s)\tilde{K}(s) \begin{bmatrix} 1 \\ 1 \end{bmatrix}, \tag{40}$$

 where the controllers $K_\beta(s)$ and $K_{\tau_g}(s)$ have to be decomposed such that:

530 $$K(s) = \begin{bmatrix} \tilde{K}_{\tau_g}(s) \\ \tilde{K}_\beta(s) \end{bmatrix} \tilde{K}(s) \tag{41}$$

where  an inverted notch filter is the outcome of combining a high-pass filter and an integrator:

$$K_{\tau_g}(s) = \tilde{K}(s)\tilde{K}_{\tau_g}(s) = \frac{2\zeta_{\tau_g}\omega_{\tau_g}\bar{k}}{s} \times \frac{s^2}{s^2 + 2\zeta_{\tau_g}\omega_{\tau_g}s + \omega_{\tau_g}^2}, \tag{42}$$

while a PI controller results from the combination of a PD and an integrator:

$$K_\beta(s) = \tilde{K}(s)\tilde{K}_\beta(s) = \frac{2\zeta_{\tau_g}\omega_{\tau_g}\bar{k}}{s} \times (\tilde{k}_p + \tilde{k}_d s), \tag{43}$$

535 where the PD controller gains are:

[Figure]

**Figure 15.** Bode plot of the normalised MISO plant $\bar{G}$ illustrating the  frequency response of each control channel separately (solid lines) as well as the response of the SISO plant $\tilde{G}$ in case of the linear combination of both actuators (dashed line) where the blade pitch actuator is active till a certain frequency before its authority deteriorates, thus, the generator torque actuator takes over from that frequency onwards.

$$\tilde{k}_p = \frac{1}{2\zeta_{\tau_g}\omega_{\tau_g}} \tag{44}$$

$$\tilde{k}_d = \frac{1}{(2\zeta_{\tau_g}\omega_{\tau_g})^2} \tag{45}$$

In this work, the damping ratio of the inverted notch ($\zeta_{\tau_g}$) is set to 0.5, and its desired natural frequency (($\omega_{\tau_g}$)) is placed at the RHPZ location of $G_{\Omega,\beta}$. The gain $\bar{k}$ in $\tilde{K}(s)$ is a static gain to either crank up or reduce the overall gain of the controllers
540    $\tilde{K}_{\tau_g}$ and $\tilde{K}_\beta$ simultaneously, and is kept at 1 in this paper. The objective is to tune one single controller instead of multiple control components, which would complicate the control tuning process.

[revised manuscript text omitted]

Pao, L. Y., Pusch, M., and Zalkind, D. S.: Control Co-Design of Wind Turbines, Annual Review of Control, Robotics, and Autonomous

730 Systems, 7, 201–226, https://doi.org/https://doi.org/10.1146/annurev-control-061423-101708, 2024.

Perez, T. and Fossen, T. I.: A matlab toolbox for parametric identification of radiation-force models of ships and offshore structures, https://doi.org/10.4173/mic.2009.1.1, 2009.

Saenz-Aguirre, A., Ulazia, A., Ibarra-Berastegi, G., and Saenz, J.: Floating wind turbine energy and fatigue loads estimation according to climate period scaled wind and waves, Energy Conversion and Management, 271, 116 303, 2022.

735 Schlipf, D., Lemmer, F., and Raach, S.: Multi-variable feedforward control for floating wind turbines using lidar, in: ISOPE International Ocean and Polar Engineering Conference, pp. ISOPE–I, ISOPE, 2020.

Skogestad, S. and Postlethwaite, I.: Multivariable feedback control: analysis and design, john Wiley & sons, 2005.

Stockhouse, D. and Pao, L. Y.: Multiloop Control of Floating Wind Turbines: Tradeoffs in performance and stability, IEEE Control Systems, 44, 63–80, https://doi.org/10.1109/MCS.2024.3432340, 2024.

740 Stockhouse, D., Phadnis, M., Henry, A., Abbas, N. J., Sinner, M., Pusch, M., and Pao, L. Y.: A Tutorial on the Control of Floating Offshore Wind Turbines: Stability Challenges and Opportunities for Power Capture, IEEE Control Systems, 44, 28–57, https://doi.org/10.1109/MCS.2024.3433208, 2024a.

Stockhouse, D., Pusch, M., Damiani, R., Sirnivas, S., and Pao, L.: Robust multi-loop control of a floating wind turbine, Wind Energy, 27, 1205–1228, https://doi.org/https://doi.org/10.1002/we.2864, 2024b.

745 van der Veen, G. J., Couchman, I. J., and Bowyer, R.: Control of floating wind turbines, in: 2012 American Control Conference (ACC), pp. 3148–3153, IEEE, https://doi.org/10.1109/ACC.2012.6315120, 2012.

WindEurope: Wind energy in Europe: 2024 Statistics and the outlook for 2025-2030, 2025.

Yu, W., Lemmer, F., Schlipf, D., Cheng, P. W., Visser, B., Links, H., Gupta, N., Dankemann, S., Counago, B., and Serna, J.: Evaluation of control methods for floating offshore wind turbines, Journal of Physics: Conference Series, 1104, 012 033, https://doi.org/10.1088/1742-

750 6596/1104/1/012033, 2018.

Yu, W., Lemmer, F., Schlipf, D., and Cheng, P. W.: Loop shaping based robust control for floating offshore wind turbines, in: Journal of Physics: Conference Series, vol. 1618, p. 022066, IOP Publishing, 2020.